# The nuclear egress complex of Epstein-Barr virus buds membranes through an oligomerization-driven mechanism

Michael K. Thorsen[1,2], Elizabeth B. Draganova[1], Ekaterina E. Heldwein[1,2]*

1 Department of Molecular Biology and Microbiology, Tufts University School of Medicine, Boston, Massachusetts, United States of America, 2 Graduate Program in Cellular, Molecular, and Developmental Biology, Graduate School of Biomedical Sciences, Tufts University School of Medicine, Boston, Massachusetts, United States of America

* katya.heldwein@tufts.edu

**Data Availability Statement:** All the data generated in this study are provided in the manuscript and the supplementary info. The coordinates and structure factors for the crystal structure of the EBV NEC

## Abstract

During replication, herpesviral capsids are translocated from the nucleus into the cytoplasm by an unusual mechanism, termed nuclear egress, that involves capsid budding at the inner nuclear membrane. This process is mediated by the viral nuclear egress complex (NEC) that deforms the membrane around the capsid. Although the NEC is essential for capsid nuclear egress across all three subfamilies of the *Herpesviridae*, most studies to date have focused on the NEC homologs from alpha- and beta- but not gammaherpesviruses. Here, we report the crystal structure of the NEC from Epstein-Barr virus (EBV), a prototypical gammaherpesvirus. The structure resembles known structures of NEC homologs yet is conformationally dynamic. We also show that purified, recombinant EBV NEC buds synthetic membranes *in vitro* and forms membrane-bound coats of unknown geometry. However, unlike other NEC homologs, EBV NEC forms dimers in the crystals instead of hexamers. The dimeric interfaces observed in the EBV NEC crystals are similar to the hexameric interfaces observed in other NEC homologs. Moreover, mutations engineered to disrupt the dimeric interface reduce budding. Putting together these data, we propose that EBV NEC-mediated budding is driven by oligomerization into membrane-bound coats.

## Author summary

Herpesviruses, which infect most of the world's population for life, translocate their capsids from the nucleus, where they are formed, into the cytoplasm, where they mature into infectious virions, by an unusual mechanism, termed nuclear egress. During nuclear budding, an early step in this process, the inner nuclear membrane is deformed around the capsid by the complex of two viral proteins termed the nuclear egress complex (NEC). The NEC is conserved across all three subfamilies of Herpesviruses and essential for nuclear egress. However, most studies to date have focused on the NEC homologs from alpha- and betaherpesviruses while less is known about the NEC from gammaherpesviruses. Here, we determined the crystal structure of the NEC from Epstein-Barr virus

have been deposited to RCSB PDB databank under accession code 7T7I.

**Funding:** This work was supported by grants from the National Institutes of Health (R01GM111795, R01AI147625 to E.E.H.; K99AI151891 to E.B.D.) and Howard Hughes Medical Institute (55108533 to E.E.H.). M.K.T. was supported by a Tufts University Graduate School of Biomedical Sciences Rosenberg Fellowship. The funders had no role in study design, data collection and analysis, decision to publish, or preparation of the manuscript.

**Competing interests:** The authors have declared that no competing interests exist.

(EBV), a prototypical gammaherpesvirus, and investigated its membrane budding properties *in vitro*. Our data show that the ability to vesiculate membranes by forming membrane-bound coats and the structure are conserved across the NEC homologs from all three subfamilies. However, the EBV NEC may employ a distinct membrane-budding mechanism due to its structural flexibility and the ability to form coats of different geometry.

## Introduction

To achieve successful replication, viruses have evolved strategies to navigate across compartmentalized eukaryotic cells. One of the more unusual mechanisms of traversing intracellular membranes is found in herpesviruses–enveloped DNA viruses that infect multiple animal species, including humans. The family *Herpesviridae*, which infect mammals, birds, and reptiles are classified into three subfamilies: alphaherpesviruses, betaherpesviruses, and gammaherpesviruses. Among them are nine herpesviruses that infect humans: herpes simplex virus types 1 and 2 (HSV-1 and HSV-2) and varicella-zoster virus (VZV) (alphaherpesviruses); human cytomegalovirus (HCMV) and human herpesvirus types 6A, B and 7 (HHV-6A/B and HHV-7) (betaherpesviruses); and Epstein–Barr virus (EBV) and Kaposi's sarcoma herpes virus (KSHV) (gammaherpesviruses). Together, these viruses infect most of the world's population for life causing a spectrum of diseases ranging from painful sores to blindness to life-threatening conditions in people with weak immune systems [1]. EBV, the focus of this work, causes infectious mononucleosis in adolescents and is associated with several hematopoietic and epithelial cell cancers [2–4] and lymphoproliferative disorders in immunocompromised patients, including those with HIV/AIDS or organ transplant recipients [5]. At present, available vaccines only target VZV while therapeutics are suboptimal. Therefore, a better knowledge of the herpesviral biology may pave the way towards effective preventives and therapeutics.

During replication, herpesviral capsids cross several host membrane barriers, traversing three distinct cellular compartments (nucleus, cytoplasm, and TGN/endosomes) while completing their assembly, before exiting the cell as infectious virions [6,7]. This process is termed egress. As the first step in egress, viral capsids, which are assembled and packaged with dsDNA genomes in the nucleus, must get across the double-membraned nuclear envelope, entering the cytoplasm to complete their maturation into infectious virions. Most cellular traffic into and out of the nucleus occurs through the nuclear pores. But the ~50-nm opening of the nuclear pore is too small to accommodate the ~125-nm capsids of herpesviruses [8]. Instead, capsids utilize a different, more complex escape route, termed nuclear egress, where capsids bud at the inner nuclear membrane (INM) and pinch off into the perinuclear space (primary envelopment). The perinuclear enveloped virions subsequently fuse with the outer nuclear membrane (ONM), releasing the capsids into the cytoplasm (de-envelopment) (reviewed in [9–11]).

Capsid budding at the INM requires the viral nuclear egress complex (NEC), a heterodimer of two conserved viral proteins: UL31, a soluble nuclear phosphoprotein, and UL34, which contains a single C-terminal transmembrane (TM) helix that anchors the NEC in the INM (reviewed in [8]). In many cases, the absence of either UL31 or UL34 causes capsids to accumulate in the nucleus, which reduces the viral titer by several orders of magnitude [12–21]. Although in rabbit skin cells infected with UL31-null HSV-1, viral titers are only moderately reduced (10- to 50-fold drop) [22], which could be due to the use of a suboptimal alternative egress route such as nuclear envelope breakdown, observed in UL34-null PRV [23].

Regardless, all studies show a reduction in viral titer in the absence of either UL31 or UL34, and thus, both genes are essential for nuclear egress across all three subfamilies of the *Herpesviridae.*

Our current understanding of the NEC function is largely based on the studies of homologs from the alphaherpesviruses HSV-1 and PRV. First, overexpression of PRV NEC in mammalian cells was found to cause formation of capsidless vesicles in the perinuclear space [24]. This demonstrated that UL31 and UL34 were the only viral proteins necessary for nuclear envelope budding. Subsequent *in-vitro* studies established the intrinsic budding ability of the NEC by showing that purified recombinant NEC from HSV-1 or PRV vesiculated synthetic lipid vesicles *in vitro* in the absence of any other factors [25,26]. The crystal structures of NEC homologs from HSV-1 [27] and PRV [27,28] provided blueprints for mechanistic studies and, in particular, revealed extensive interactions that stabilize the heterodimer. Finally, cryogenic electron microscopy and tomography (cryo-EM/ET) studies showed that the NEC oligomerizes into hexagonal coats on the inner surface of budded vesicles formed by recombinant HSV-1 NEC *in vitro* [25,43], in uninfected cells overexpressing PRV NEC [29], and in perinuclear enveloped vesicles purified from HSV-1-infected cells [30]. A high-resolution view of the NEC/NEC oligomeric interfaces was afforded by the crystal structure of HSV-1 NEC [27], in which the NEC formed a hexagonal lattice with the same geometry and dimensions as the hexagonal membrane-bound coats formed during budding [25,29,30,43]. Follow-up work confirmed that HSV-1 NEC mutations that disrupt oligomeric interfaces reduce budding *in vitro* [27] and in infected cells [31,32]. Collectively, these findings established the NEC of alphaherpesviruses as a robust membrane-budding machine that forms hexagonal membrane-bound coats (reviewed in [8]).

While less well studied than the alphaherpesvirus homologs, the NECs from the betaherpesviruses HCMV and murine cytomegalovirus (MCMV) are also essential for successful viral replication [33,34]. Insertions or deletions within the conserved regions of MCMV UL53, a UL31 homolog, reduce viral replication [35,36]. Additionally, point mutations in HCMV UL50, a UL34 homolog, designed to interfere with NEC formation, also reduced viral replication, thereby underscoring the importance of complex formation for functionality [37]. The crystal structures of the NEC homolog from the betaherpesvirus HCMV [38,39] revealed structural similarities with the NEC from alphaherpesviruses HSV-1 [27] and PRV [27,28] despite the relatively low sequence identity. Moreover, in one of the structures, the HCMV NEC [39] formed a hexagonal lattice with the same geometry and dimensions as those formed by the HSV-1 [25,27,30,43] and PRV counterparts [29]. Altogether, this implied a common NEC-mediated mechanism of primary capsid envelopment between the alpha- and betaherpesviruses.

In gammaherpesviruses EBV and KSHV, mechanistic studies of the NEC function utilized overexpression in mammalian cells or insect cells infected with recombinant baculovirus, respectively [40,41]. Overexpression of UL34 homologs, EBV BFRF1 [41] or KSHV ORF67 [40], caused formation of proliferations of nuclear and cytoplasmic membranes that resemble stacks or tubules. In contrast, overexpression of both UL34 and UL31 homologs (EBV BFRF1 and BFLF2, KSHV ORF67 and ORF69) produced curved multilayered cisternae at the INM in the case of EBV [41] and perinuclear vesicles in the case of KSHV [40], the latter resembling the perinuclear vesicles observed in mammalian cells overexpressing PRV NEC [24]. These findings implicated a conserved role for the gammaherpesviral NEC in nuclear envelope remodeling. However, neither the intrinsic budding ability nor coat formation have yet been demonstrated for NEC from any gammaherpesvirus. Moreover, the available structural information is limited to EBV BFRF1 bound to a short segment of BFLF2 [42]. To fill in these knowledge gaps, we pursued structural and functional studies of EBV NEC.

Here, we show that the purified, recombinant EBV NEC vesiculates synthetic lipid membranes *in vitro* and forms membrane-bound coats, which suggests that the intrinsic membrane budding ability is a conserved property of the NECs across *Herpesviridae*. We also report the most complete crystal structure of EBV NEC to date and show that it resembles known structures of NEC homologs from the other two subfamilies. Importantly, EBV NEC crystals contain five independent, structurally distinct heterodimers in the asymmetric unit. The structural differences among the 5 EBV NEC heterodimers, notably within BFLF2 and at the BFLF2/BFRF1 interface, for the first time experimentally demonstrate conformational dynamics within the EBV NEC. However, instead of hexamers, the EBV NEC forms dimers in the crystals, and its membrane-bound coats formed *in vitro* appear different from the hexagonal geometry observed in NEC coats of alphaherpesviruses [25,29,43]. The dimeric interfaces observed in the EBV NEC crystals are similar to the hexameric interfaces observed in other NEC homologs. Moreover, mutations engineered to disrupt dimeric interfaces reduce budding. Therefore, we propose that while the NEC operates as an oligomerization-driven membrane budding machine, the EBV NEC coats have a different, yet unknown geometry, potentially, due to the structural flexibility of the EBV NEC.

## Results

### EBV NEC crystallization and structure determination

For crystallization, we designed a construct EBV NEC195Δ65 composed of residues 1–195 of BFRF1 and residues 66–318 of BFLF2 (Fig 1A), which is similar to those of previously crystallized NEC homologs [27,28,38,39,42]. This construct lacks the membrane-proximal regions (MPRs) in both BFRF1 and BFLF2 as well as the TM anchor of BFRF1 (Fig 1A). The EBV NEC195Δ65 was expressed in *E. coli* and purified using affinity and size-exclusion chromatography, similarly to HSV-1 and PRV NEC [27]. Crystals were grown by vapor diffusion in hanging drops, and unusually, they appeared in drops that contained only the protein complex but no reservoir solution and were equilibrated over the reservoir solution (25% PEG 3350, 0.1 M Tris-HCl pH 8.5, 0.2 M $Li_2SO_4$). The crystals took the $P4_32_12$ space group with five NEC heterodimers in the asymmetric unit, termed NEC1-5. The structure was determined by a combination of single anomalous dispersion of the $Zn^{2+}$ ion and molecular replacement using the structure of BFRF1 fused to the BFLF2 hook by a flexible linker (rcsb pdb 6t3z [42]) as the search model. The structure was refined to 3.97 Å resolution ($R_{work}$ = 27.0%, $R_{free}$ = 30.8%) (Table 1). Atomic coordinates and structure factors for the EBV NEC structure have been deposited to the RCSB Protein Data Bank under accession number 7t7i.

### Overall structure of the EBV NEC

Like its counterparts in alpha- and betaherpesviruses, the EBV NEC resembles an elongated cylinder that is composed of the globular BFRF1 pedestal topped with the globular C-terminal domain of BFLF2 and the N-terminal two-helix "hook" of BFLF2 that wraps around the BFRF1 pedestal (Fig 1B). There are five EBV NEC heterodimers in the asymmetric unit, NEC1-5, composed of BFRF1 chains A, C, E, G, and I and BFLF2 chains B, D, F, H, and J (Fig 1B). Three heterodimers, NEC1 (A/B), NEC2 (C/D), and NEC4 (G/H), as well as BFRF1 of NEC3 (E) are well resolved (89–99% of all residues) whereas NEC5 (I/J) and BFLF2 of NEC3 (F) are less well resolved (75–78% of all residues) (S1 Table). The NEC heterodimers have notable structural differences (Fig 1C–1F), mainly in BFLF2 and the relative orientations of BFRF1 and BFLF2 as described in detail below. They can be superimposed with the root mean square deviations (RMSDs) ranging from 1.16 Å to 2.57 Å (S2 Table).

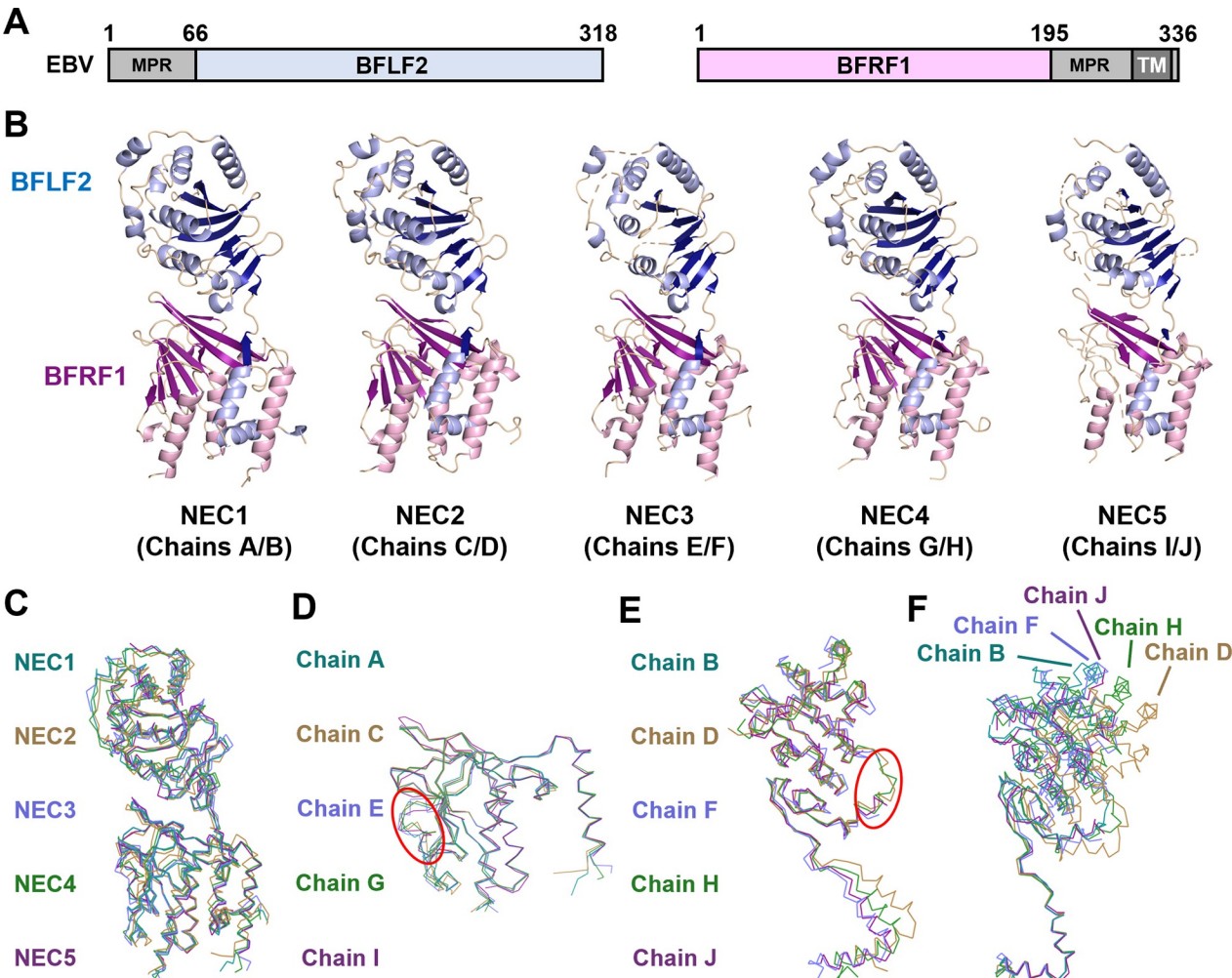

**Fig 1. Crystal structure of EBV NEC.** (A) Schematic depicting construct boundaries for EBV NEC (rcsb pdb 7t7i). BFLF2 is shown in blue while BFRF1 is shown in pink. Grey regions denote residues omitted from constructs including the membrane-proximal regions (MPRs) and the transmembrane (TM) region in darker grey. (B) Crystal structures of the 5 EBV NEC copies in the asymmetric unit with chain IDs. BFLF2 is shown in blue with light blue for α-helices and dark blue for β-strands. BFRF1 is shown in pink with light pink for α-helices and dark pink for β-strands. (C) Ribbon overlay aligned to NEC2 (Chains C+D) for the 5 copies of EBV NEC in the asymmetric unit. NEC1 (teal), NEC2 (gold), NEC3 (light purple), NEC4 (green), and NEC5 (purple). (D) Ribbon overlay aligned to chain A for the 5 copies of BFRF1 in the asymmetric unit. Red oval highlights differences in β5/β6 loop. Chain A (teal), chain C (gold), chain E (light purple), chain G (green), and chain I (purple). (E) Ribbon overlay aligned to the globular domain of B for the 5 copies of BFLF2 in the asymmetric unit. Red oval highlights differences in β2/β3 loop. Chain B (teal), chain D (gold), chain F (light purple), chain H (green), and chain J (purple). (F) Ribbon overlay of the 5 copies of BFLF2 aligned to BFRF1 chain A (hidden) in the asymmetric unit. Same coloring as (E). Images were created in PyMol [44].

## The BFRF1 structure

The BFRF1 structure is similar to those of the UL34 homologs in alpha- and betaherpesviruses and is composed of two β-sheets arranged into a novel fold of a splayed β-sandwich, previously termed a β-taco [37], decorated with four α-helices (S1A Fig). Helices α1, α2, and α4 are oriented parallel to each other and to the longest axis of the NEC and brace helix α2 of BFLF2 hook. In chains A, C, E, and G, all residues except the last C-terminal residue 195, are resolved. These four chains can be superimposed with RMSDs ranging from 0.72 to 0.79 Å (S2 Table). Several loops in these four chains, notably, the β5/β6 loop, adopt different conformations (Fig 1D). Additionally, chain E contains the longest α4 helix but lacks the short β6 strand (S2A

**Table 1. Data collection and refinement statistics.**

| Parameter | Value[a] |
|---|---|
| **Data collection statistics** | |
| Wavelength | 1.28215 |
| Space group | P4$_3$2$_1$2 |
| Unit cell (Å, °) | a = b = 238.153, c = 137.532 <br> α = β = γ = 90 |
| Resolution range (Å) | 168.54–3.97 (4.19–3.97) |
| *No. of reflections* | |
| Total | 518,063 (78,362) |
| Unique | 34,632 (4,965) |
| Multiplicity | 15.0 (15.8) |
| Completeness (%) | 100 (100) |
| Mean $I/\sigma(I)$ | 10.3 (1.2) |
| Wilson B-factor (Å$^2$) | 143.375 |
| $R_{merge}$[b] | 0.285 (3.213) |
| $R_{meas}$[b] | 0.305 (3.403) |
| $R_{pim}$ | 0.079 (0.852) |
| $CC_{1/2}$ | 0.991 (0.430) |
| **Refinement statistics** | |
| *No. of reflections used*: | |
| In refinement | 34,522 (3,265) |
| For $R_{free}$ | 2,000 (189) |
| $R_{work}$ | 0.2699 (0.3656) |
| $R_{free}$ | 0.3081 (0.3937) |
| $Cc_{work}$ | 0.898 (0.644) |
| $CC_{free}$ | 0.886 (0.704) |
| *No. of*: | |
| Nonhydrogen atoms | 16,142 |
| Macromolecules | 16,134 |
| Ligands | 5 |
| Solvent | 3 |
| Protein residues | 2,062 |
| *RMSD* | |
| Bond length (Å) | 0.003 |
| Bond angle (°) | 0.75 |
| *Ramachandran plot (%)* | |
| Favored regions | 92.61 |
| Allowed regions | 6.29 |
| Outliers | 1.10 |
| *Rotamer outliers (%)* | 3.07 |
| *Clash score* | 7.581 |
| *B-factor* | |
| Avg | 200.07 |
| Macromolecules | 200.08 |
| Ligands | 156.53 |
| Solvent | 199.87 |

[a]Statistics for the highest-resolution shell are shown in parentheses.

[b]$R_{work}$ and $R_{free}$ are defined as $\Sigma||F_{obs}|-|F_{calc}||/\Sigma|F_{obs}|$ for the reflections in the working or the test set, respectively.

[c]As determined using MolProbity (molprobity.biochem.duke.edu) [45].

Fig). Chain I has several unresolved regions and has the largest RMSDs, 1.01 to 1.14 Å
(S2 Table).

## The BFLF2 structure

BFLF2 structure resembles those of the UL31 homologs in alpha- and betaherpesviruses and is
composed of a globular domain and a helical hook. The globular domain of BFLF2 consists of
two β-sheets decorated with a helical cap plus several additional helices (S1B Fig). The con-
served CCCH-type Zn-binding site, formed by three cysteines and one histidine, is located at
the base of the globular domain. Chains B, D, and H are well resolved and lack only the N ter-
mini (residues 66–77, 66–76, and 66–77, respectively); chain D also lacks the C terminus, resi-
dues 317–318. In contrast, chains F and J have several unresolved regions (S2B Fig). The five
BFLF2 copies have several structural differences. First, several loops, such as the β2/β3 loop,
adopt different conformations (Fig 1E). Further, depending on the BFLF2 chain analyzed, a
structural element may be missing, or its length may differ. For example, chain F is missing the
β3 strand that is found in chains B, D, H, and J whereas chains B, D, F have a longer β1 (S2B
Fig). Finally, chain D has a short $3_{10}$ helix preceding helix α1 (S2B Fig).

   The most notable difference, however, is the distinct relative orientations of the globular
domain and the hook in the five BFLF2 copies (Fig 1E and 1F) that result in somewhat differ-
ent BFLF2/BFRF1 interfaces across the five EBV heterodimers as described below. Due to
these conformational differences, BFLF2 chains superpose onto one another with RMSDs
ranging from 1.25 Å to 2.41 Å (S2 Table).

## Structural comparisons of the NEC homologs

Despite the modest sequence identity (S3 Table), the EBV NEC structure closely resembles the
homologs from HCMV, HSV-1, and PRV (Fig 2A, 2B and 2C), with the overall RMSDs rang-
ing from 2.64 Å to 3.83 Å depending on the EBV NEC heterodimer (S4 Table). The BFRF1
structure is more similar to HCMV UL50 than to HSV-1 or PRV UL34 (S4 Table). Most struc-
tural elements superpose well (Fig 2D), but several helices vary in length, with helices α1, α2,
and α4 being the longest in EBV (S3A Fig). Additionally, several loops are of different length
and adopt different orientations (loops α1/β1, β1/β2, β2/β3, α2/β4, β5/β6, α3/β9, and β9/β10)
(Figs 2D and S3A), with loop β1/β2 containing a short helix in EBV and HCMV but not in
HSV-1 or PRV (S3A Fig). The BFLF2 chains largely contain the same structural elements as
HSV-1 and PRV UL31 and HCMV UL53 (S3B Fig), with some differences. For example, the
EBV BFLF2 chains lack the short helix after β3 in HSV-1 and PRV UL31 (S3B Fig). Overall,
EBV NEC heterodimers are somewhat more similar to each other (S2 Table) than to other
homologs (S4 Table).

## The BFRF1/BFLF2 interactions and conformational dynamics in the EBV NEC

BFLF2 interacts with BFRF1 at two interfaces, the so-called "primary" or hook-in-groove
interface that involves the BFLF2 hook, and the "secondary" or globular interface, which
involves the globular domain of BFLF2. The hook-in-groove interface was previously visual-
ized in the structure of BFRF1 fused to the BFLF2 hook by a flexible linker with an interface of
1,569 Å[2] and described in detail [42]. Therefore, we will not elaborate on it except to note that
the interface varies from 1,403 to 1,727 Å[2] across the 5 heterodimers (S2A and S2B Fig and S5
Table) and contributes roughly 80% of the total BFRF1/BFLF2 interface. The differences in the
area buried at the hook-and-groove interface are due mainly to the number of interacting

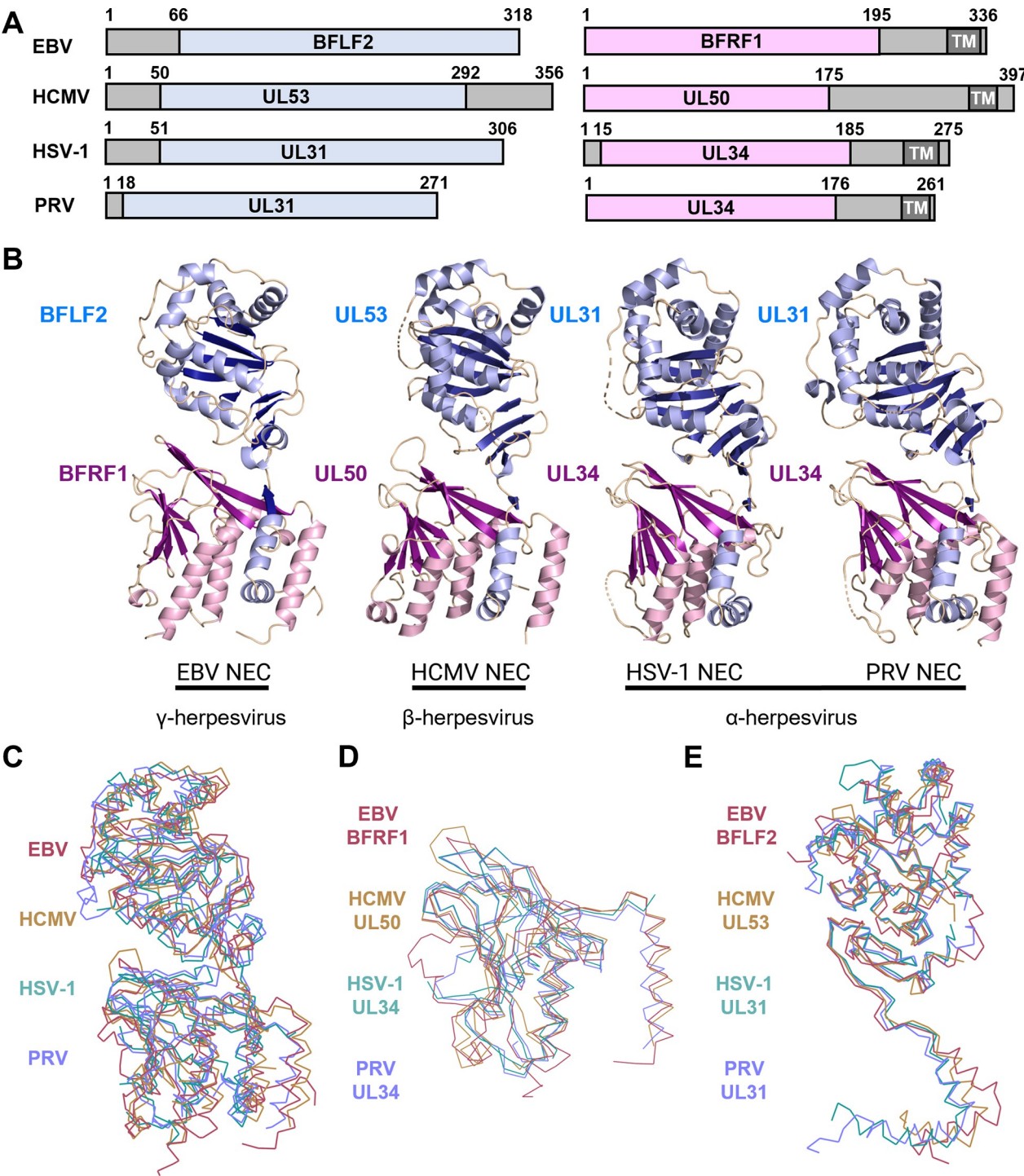

**Fig 2. Crystal structure of EBV NEC and homologs.** (A) Schematic depicting construct boundaries for the NECs from EBV (rcsb pdb 7t7i), HCMV (rcsb pdb 5d5n), HSV-1 (rcsb pdb 4zxs) and PRV (rcsb pdb 4z3u). BFLF2 and homologs are shown in blue while BFRF1 and homologs are shown in pink. Grey regions denote residues omitted from constructs with the transmembrane (TM) region in darker grey. (B) Crystal structures of homologous NECs depicted in (A). BFLF2 and homologs are shown in blue with light blue for α-helices and dark blue for β-strands. BFRF1 and homologs are shown in pink with light pink for α-helices and dark pink for β-strands. EBV NEC chains C+D are shown. (C) Ribbon overlay aligned to NEC2 (Chains C+D) for NEC homologs. EBV (dark red), HCMV (gold), HSV-1 (light teal), PRV (light purple) (D) Ribbon overlay aligned to EBV chain C for the BFRF1 homologs. Same coloring as (C). (E) Ribbon overlay aligned to EBV chain D for the BFLF2 homologs. Same coloring as (C). Images created in PyMol [44] and BioRender.com.

residues (S2A and S2B Fig) and minimally contribute to the structural plasticity of the EBV NEC described below.

By contrast, the globular domain interface contributes approximately 20% of the total BFRF1/BFLF2 interface, burying 333 to 464 $\text{Å}^2$ across the 5 NEC copies (S5 Table). Across the five EBV NEC heterodimers, the globular domains of BFLF2 adopt distinct orientations relative to BFRF1 (Fig 3A) where they are either more or less tilted towards the BFRF1/hook module, as measured by the angles between $D10_{BFRF1}/P98_{BFRF1}/D287_{BFLF2}$. The tilt towards the hook increases, with a concomitant decrease in the angle in B<F<H<D (104°>100°>94°>89°) (Fig 3A). Chain I could not be analyzed due to $D287_{BFLF2}$ being unresolved, but the overall alignment is very similar to chains B and F (Fig 3A). Major interactions are polar and include a salt bridge $D115_{BFRF1}$-$R128_{BFLF2}$, observed in all NEC copies except in the poorly resolved NEC5, and a hydrogen bond between $T158_{BFRF1}$ and $E250_{BFLF2}$ in NEC1 and NEC4 (Fig 3B).

We attribute the differences in tilt across the EBV NEC heterodimers to the differences in the interactions at the globular domain interface, specifically, the presence or the absence of the distal hydrogen bond between $T158_{BFRF1}$ and $E250_{BFLF2}$ (Fig 3B). We hypothesize that in NEC1, this hydrogen bond keeps BFLF2 anchored to BFRF1 and in an upright position (Fig 3B). By contrast, in NEC2, the lack of the hydrogen bond (Fig 3B) allows the globular domain of BFLF2 to tilt away from the BFLF2/BFRF1 interface and towards the BFLF2 hook. This tilted orientation is stabilized by the crystal contacts between chain D and chain H of an NEC symmetry mate.

Analysis of the heterodimeric interfaces revealed no conserved identical residues across EBV NEC and its homologs.

## NEC/NEC interactions

Oligomerization of the NEC into the hexagonal "honeycomb" lattice is a major driving force of membrane budding (reviewed in [8]). This lattice has been visualized in the cryo-ET reconstructions of the HSV-1 NEC coats formed *in vitro* [25,43] and in infected cells [30] as well as in PRV NEC coats formed in cells overexpressing the NEC [29]. Additionally, this lattice was also observed in HSV-1 NEC and HCMV NEC crystals (Fig 4A and 4B). Structure-guided mutagenesis of hexameric and inter-hexameric interfaces in the HSV-1 NEC hexagonal lattice established its significance for budding *in vitro* [25,27] and nuclear egress in infected cells [31,32].

The EBV NEC did not form hexamers in the crystals, however. Instead, four EBV NEC copies formed two dimers, NEC1/NEC4 and NEC2/NEC3 (Fig 4C and 4D). The dimeric interfaces in EBV NEC crystals are very similar to the hexameric interfaces in the crystals of HSV-1 and HCMV NEC, and when comparing them, we refer to them henceforth as oligomeric. The dimeric interfaces in EBV NEC are larger (759 $\text{Å}^2$ for NEC1/NEC4 and 967 $\text{Å}^2$ for NEC2/NEC3) than the hexameric interfaces in HSV-1 NEC (613 $\text{Å}^2$ for $NEC_{AB}$ and 572 $\text{Å}^2$ for $NEC_{CD}$) and HCMV NEC (760 $\text{Å}^2$) (S6 Table). This could be due to a somewhat larger number of residues at the N termini of both BFRF1 and BFLF2 that contribute to the interface (Fig 5 and S6 Table). Analysis of the oligomeric interfaces revealed only a slight enrichment for the BFLF2 interface of NEC2/NEC3 and NEC1/NEC4 (S7 Table), but no enrichment for BFRF1, possibly, because the interfaces are mainly formed by similar rather than identical residues.

## EBV NEC has an intrinsic membrane budding activity

To measure the membrane budding activity of the EBV NEC, we generated and purified a recombinant soluble construct NEC228Δ15-His$_8$ lacking the first 15 residues of BFLF2 and residues 229–336 of BFRF1, which eliminated the transmembrane domain. In its place, a His$_8$-

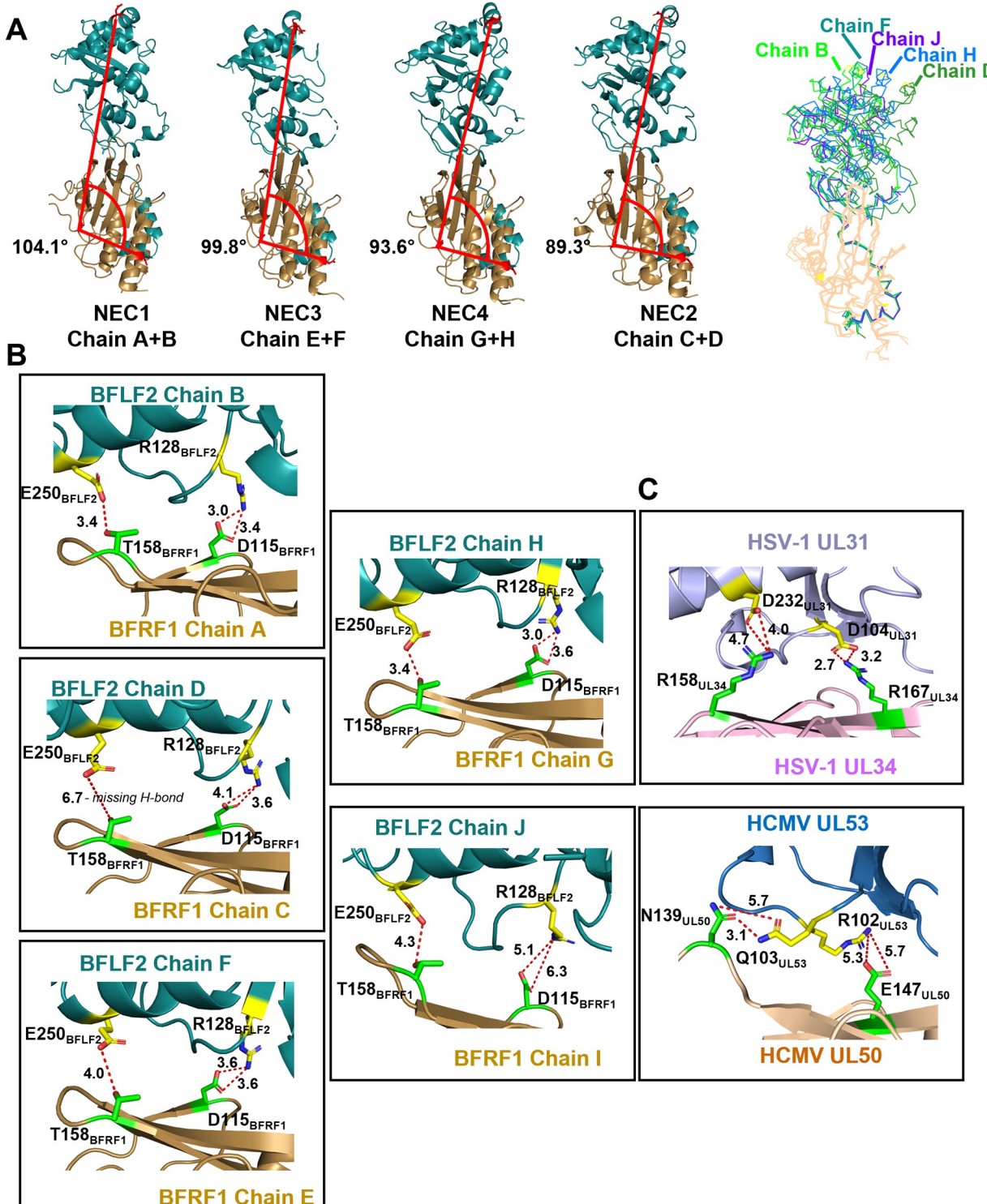

**Fig 3. BFLF2 adopts different tilt angles relative to BFRF1.** (A) Crystal structures of EBV NEC heterodimers, arranged in the order from those with the largest tilt angle to those with the smallest tilt angle. Tilt angle (red line) was measured from the Cα atoms of $D10_{BFRF1}$/$P98_{BFRF1}$/$D287_{BFLF2}$. $D287_{BFFL2}$ was unresolved in chain J, and thus the tilt angle could not be determined for NEC5 (chains I/J). BFLF2 (teal) and BFRF1 (tan). The last image shows an overlay of all 5 NECs with BFLF2 chain B (neon green), chain D (green), chain F (teal), chain H (blue), chain J (purple) and BFRF1 (tan). (B) Close-up view of the globular domain interactions between BFLF2 (teal) and BFRF1 (tan). Polar contact residues for BFLF2 (yellow) and BFRF1 (green) are shown with measurements (Å). (C) Close-up view of the globular domain interactions between HSV-1 UL31 (light purple) and UL34 (light pink) vs. HCMV UL53 (blue) and UL50 (tan). Polar contact residues for UL31/UL53 (yellow) and UL34/

UL50 (green) are shown with measurements (Å). Crystal structures of the NEC homologs from HSV-1 (rcsb pdb 4zxs) and HCMV (rcsb pdb 5d5n) were used. All images were generated in PyMol [44].

tag was added to the C terminus of BFRF1 (Fig 6A) to tether the complex to membranes. Histidine tags–when used in conjunction with nickel-chelating lipids–act as membrane anchors [49]. Using the established *in-vitro* budding assay utilizing giant unilamellar vesicles (GUVs) (Fig 6B and 6C) [25,50,51], we observed that EBV NEC228Δ15-His$_8$ (Fig 6A) mediated budding at a level comparable to that of HSV-1 NEC220-His$_8$ (Fig 6D). Therefore, we conclude that the intrinsic membrane budding ability previously reported for alphaherpesviruses HSV-1 and PRV is conserved in EBV, a gammaherpesvirus.

In HSV-1 and PRV, the N-terminal membrane-proximal region (MPR) of UL31 (residues 1–50 in HSV-1), specifically, clusters of basic residues, are required for membrane budding *in vitro* [50] and nuclear egress in infected cells [52], respectively. To probe the importance of the putative MPR of EBV BFLF2 (residues 1–65) in budding, we generated a series of N-terminal truncations, EBV NEC228Δ24-His$_8$, EBV NEC228Δ34-His$_8$, and EBV NEC228Δ44-His$_8$ (Fig 6A). We found, however, that only the longest truncated construct, EBV NEC228Δ15-His$_8$, efficiently budded membranes (15%) whereas the rest, e.g., EBV NEC228Δ24-His$_8$, did not (Fig 6D). Therefore, residues 16–24 of BFLF2 are required for budding. The sole basic cluster in the putative BFLF2 MPR, residues R22/R23, is located within this region. We hypothesize that, by analogy with basic clusters in HSV-1 UL31 MPR [50], this dibasic motif interacts with membranes and is essential for the EBV NEC budding activity.

## EBV NEC forms membrane-bound coats inside budded vesicles

The HSV-1 NEC oligomerizes into hexagonal coats during budding events *in vitro* [25,43] and in infected cells [30]. To investigate if the EBV NEC also forms coats during *in-vitro* budding, we used EBV NEC215-N31S$_{BFLF2}$ side by side with HSV-1 NEC220 in membrane-budding experiments and imaged them using cryo-EM. HSV-1 NEC220 is the construct used in prior *in-vitro* budding experiments and oligomerizes into a hexagonal coat [25,43]. EBV NEC215-N31S$_{BFLF2}$ is the construct analogous to HSV-1 NEC220 based on secondary structure alignment (S3 Fig) that contains a N31S mutation in BFLF2 introduced to reduce spontaneous proteolytic cleavage of BFLF2 during expression and purification and improve protein yield.

Each protein complex was incubated with large unilamellar vesicles (LUVs) of size and composition previously used for cryo-EM visualization of HSV-1 NEC220 budding [25]. In both EBV and HSV-1 NEC samples, we observed vesicles with thickened membranes containing internal membrane-bound coats (Fig 7A–7D). In contrast, non-budded LUVs had thinner membranes (Fig 7A and 7D). Both the EBV and HSV-1 NEC coats are ~11-nm thick in side views (Fig 7B and 7D insets; side view). Our previous cryo-EM experiments with HSV-1 NEC [25,43] established that these coats are composed of a single membrane-bound NEC layer. Therefore, based on our measurements, we hypothesize that the coats formed in the presence of EBV NEC are also composed of a single membrane-bound EBV NEC layer. In both cases, instead of ILVs inside "mother" vesicles, only individual budded vesicles were observed, similarly to the original studies with HSV-1 NEC [25]. We hypothesize that the excess amount of NEC in the cryo-EM experiments resulted in multiple rounds of budding such that the membrane of the "mother" vesicle was depleted, releasing budded vesicles containing internal NEC coats.

Although current resolution is insufficient to discern the precise oligomeric arrangement of the HSV-1 NEC coats, we hypothesize that the HSV-1 NEC coats visualized here have

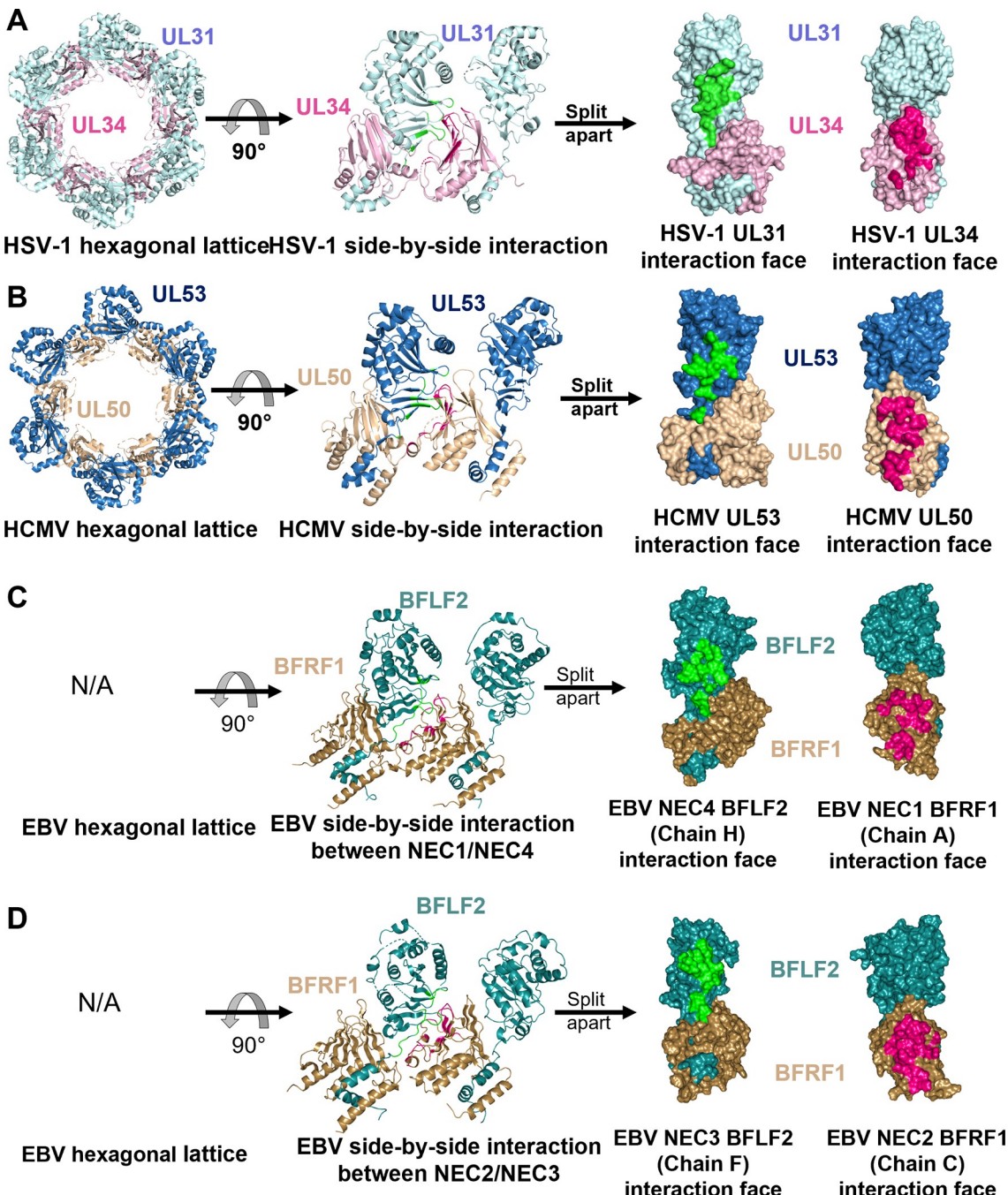

**Fig 4. Oligomeric interfaces formed by HSV-1, HCMV, and EBV NEC homologs in crystals.** Hexameric interfaces in crystals of HSV-1 NEC (rcsb pdb 4zxs) (A) and HCMV NEC (rcsb pdb 5d5n) (B), as well as dimeric interfaces in two EBV NEC heterodimer pairs (rcsb pdb 7t7i, this study) (C and D). Left column: NEC hexamers observed in the crystals of HSV-1 and HCMV are shown as top view. Center column: the NEC/NEC interfaces in HSV-1 and HCMV and EBV homologs are shown as side view. Right column: the same NEC/NEC interfaces are split, rotated towards the viewer, and shown in surface representation. HSV-1 UL31 (light blue), HSV-1 UL34 (pink), HCMV UL53 (blue), HCMV UL50 (beige), EBV BFLF2 (teal), EBV BFRF1 (brown). Interface residues were assigned by PDBePISA [46]. Only residues with 30% or more of their surface area buried at the oligomeric interface are shown, in green and mauve in UL31 and UL34 homologs, respectively.

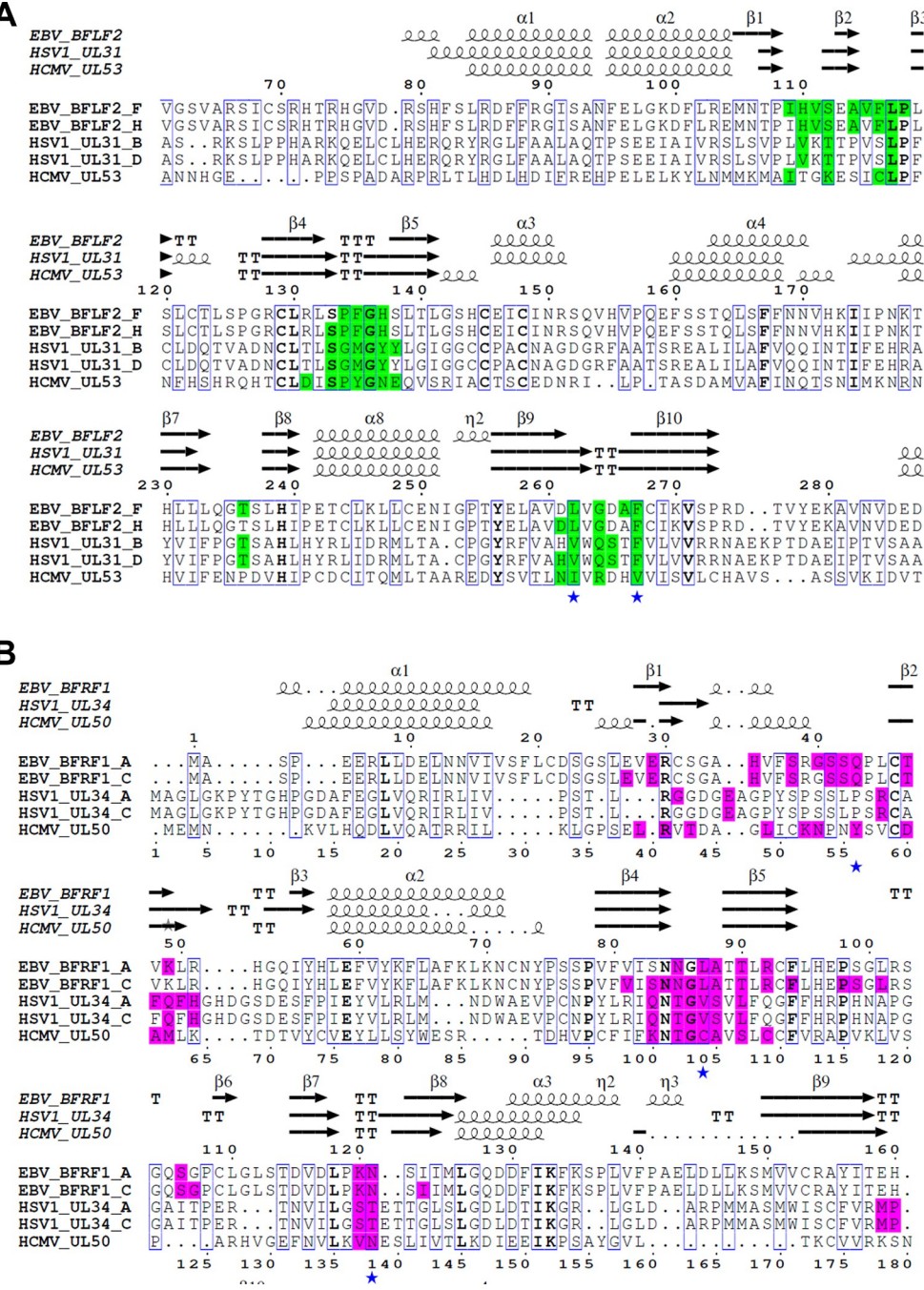

**Fig 5. Sequence alignment of regions mediating NEC oligomerization.** (A) Sequence alignment of UL31 homologs, EBV BFLF2, HSV-1 UL31, and HCMV UL53. (B) Sequence alignment of UL34 homologs, EBV BFRF1, HSV-1 UL34, and HCMV UL50. For HSV-1 and EBV NEC, two chains are shown for each NEC component because two distinct oligomeric interfaces were observed in crystals of both HSV-1 and EBV NEC. For simplicity, only regions containing the interface residues are shown. Alignment was made in Clustal Omega [47] and annotated using Espript [48]. Interface residues were assigned by PDBePISA [46]. Residues with 30% or more of their surface area buried at the oligomeric interface are highlighted in green and mauve in UL31 and UL34 homologs, respectively. Interface residues mutated in this study are marked with a blue star below the sequence.

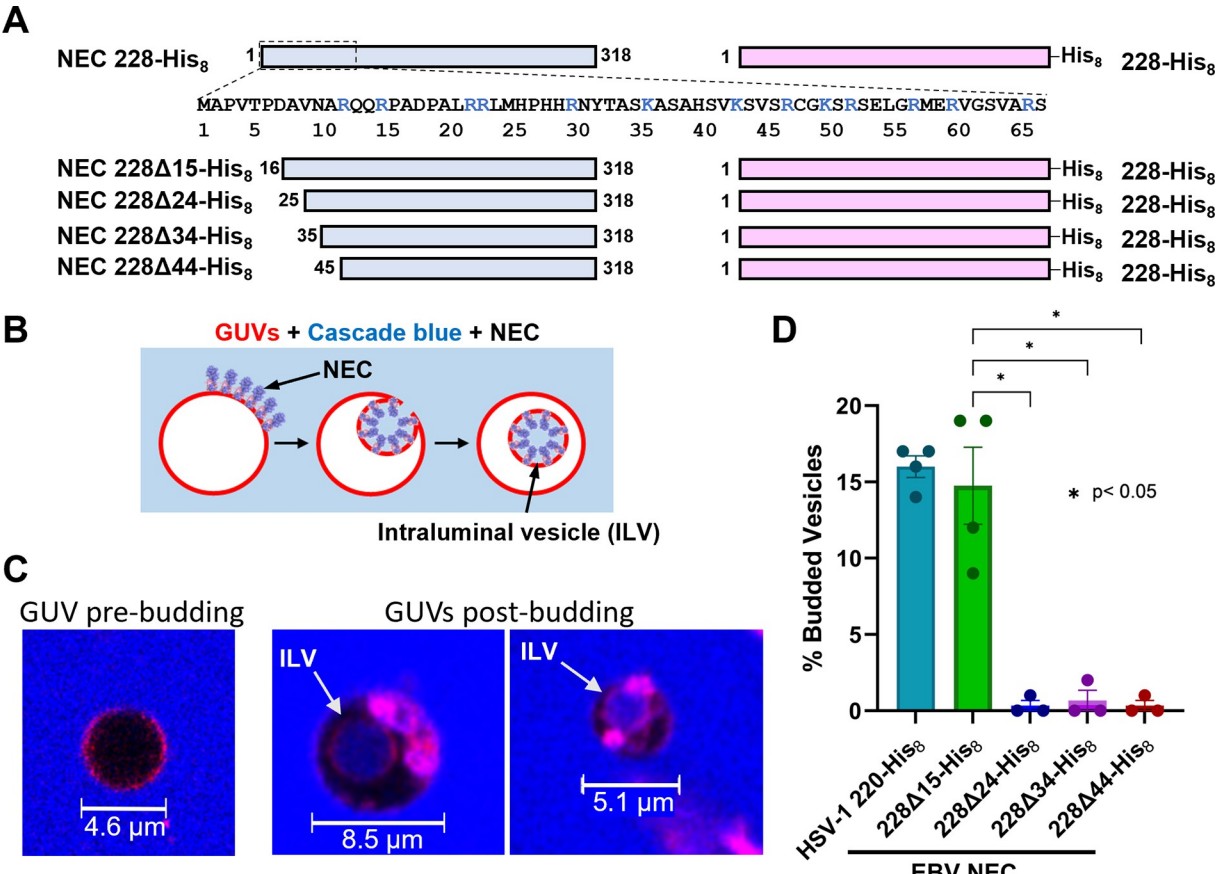

**Fig 6. EBV NEC buds synthetic membranes.** (A) EBV NEC constructs used. Sequence of EBV BFLF2 1–67 are shown. (B) A diagram of the in-vitro budding assay. GUVs (red) bind NEC (purple and pink) which generates formation of membrane curvature allowing cascade blue (light blue) to enter the forming ILV. (C) Images of pre and post NEC-budded vesicles from confocal experiments. ILVs are shown with arrows. (D) In-vitro budding assay. Vesicles contain Ni-chelating lipids to tether His$_8$-tagged NEC to membranes. % budding was determined by counting the number of intraluminal vesicles (ILVs) after addition of NEC. Background levels of ILVs in the absence of NEC, typically around 10%, were subtracted from all values. Significance to 228Δ15-His$_8$ was calculated using an unpaired Student's t-test with Welch's correction (p<0.05 = *). In all plots, error bars represent the standard error of the mean (68% confidence interval of the mean) for at least three individual experiments.

hexagonal geometry because their appearance closely resembles the coats formed by the same HSV-1 NEC construct that had hexagonal geometry revealed by 3D averaging [25,43]. In contrast, the top views of the EBV NEC coats (Fig 7B inset, top view) did not resemble the top views of HSV-1 NEC coats (Fig 7D inset, top view). Consequently, the differences in appearance between the HSV-1 and EBV NEC coats leads us to speculate that the EBV NEC oligomerizes into coats of a different geometry, possibly, due to the structural plasticity observed in the EBV NEC crystal structure. Such conformational flexibility may influence the interactions between neighboring NECs enabling the assembly of an alternative NEC lattice. The detailed analysis of the EBV NEC coat geometry is the subject of future work.

## The NEC oligomeric interface is important for budding

The dimeric NEC/NEC interfaces observed in EBV crystals closely resemble the hexameric NEC/NEC interfaces observed in crystals of HSV-1 and HCMV NEC. Mutations engineered to disrupt hexameric interfaces in HSV-1 NEC reduce budding *in vitro* [25,27] and nuclear

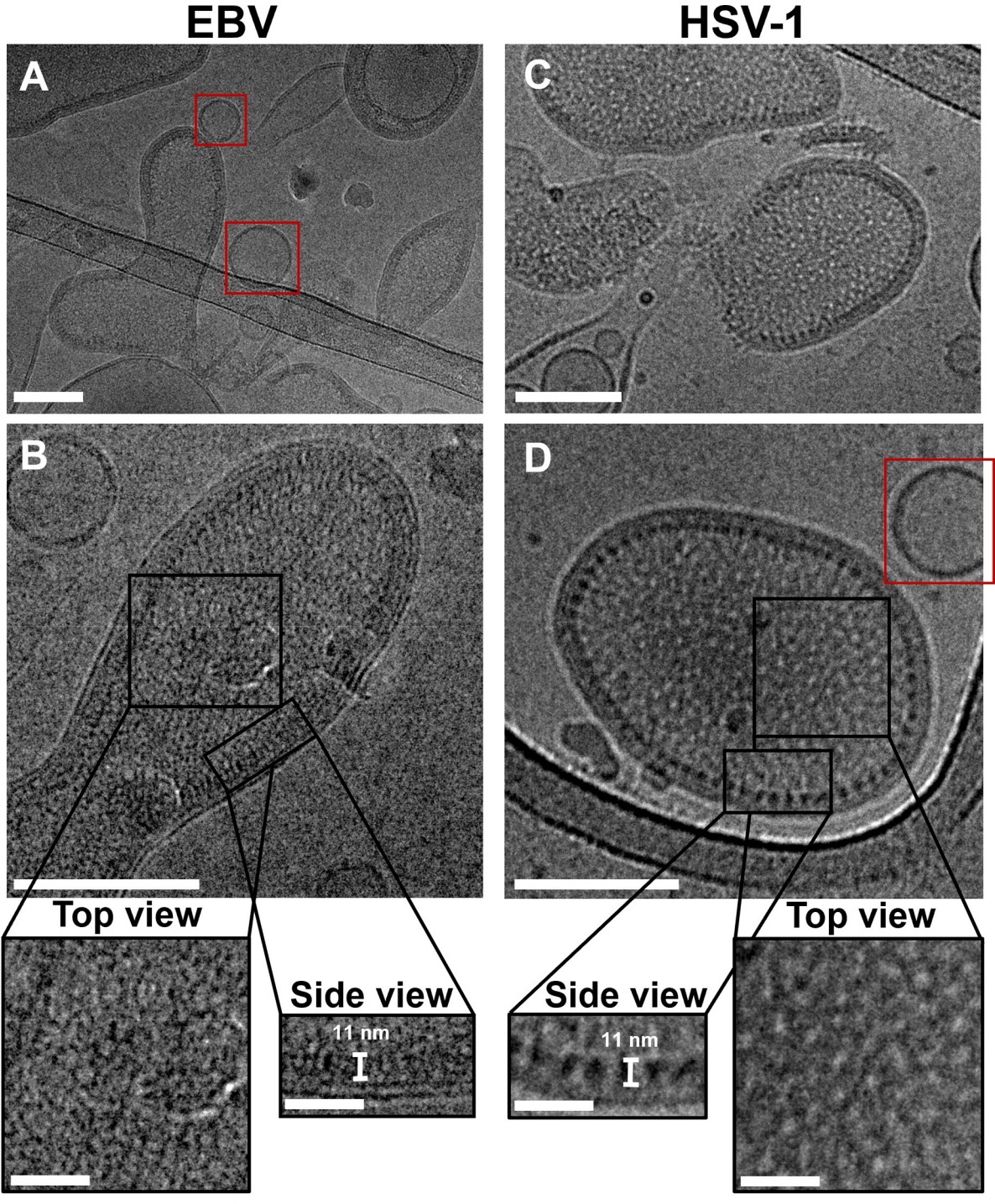

**Fig 7. EBV and HSV-1 NEC form internalized membrane-bound coats on liposomes.** Cryo-EM shows that both EBV NEC215-N31S$_{BFLF2}$ (A, B) and HSV-1 NEC220 (C, D) form internal membrane-bound coats. Both NECs are ~11 nm in height, as previously determined for HSV-1 NEC. HSV-1 NEC forms the canonical coats (B inset; top views) also seen by our group in previous publications [25,43] while the oligomeric arrangement of the EBV NEC coats is yet unclear. Red boxes denote LUVs not bound or budded by either NEC with thinner membranes. Scale bars = 100 nm. Inset scale bars = 25 nm. Insets are shown on the same scale.

egress in infected cells [31,32], confirming the significance of oligomerization for membrane budding.

To probe the importance of the dimeric interface in EBV NEC budding, we mutated interface residues equivalent to those that form hexameric interfaces in HSV-1 NEC. Prior mutational analysis of the hexameric interface in HSV-1 NEC pinpointed the hydrophobic triad ($V92_{UL34}$/$V247_{UL31}$/$F252_{UL31}$) and a polar residue ($T123_{UL34}$) near the top margin of the interface along with a polar residue $E37_{UL34}$ near the bottom margin of the interface (Fig 8A), which forms a hydrogen bond with $T89_{UL31}$ of the neighboring NEC, as important for budding *in vitro* [25,27] and in infected cells [31,32]. Guided by these mutations, we mutated the equivalent hydrophobic "triad" residues in EBV, $L87_{BFRF1}$, $L262_{BFLF2}$, and $F267_{BFLF2}$, as well as the polar $N121_{BFRF1}$, the equivalent of $T123_{UL34}$ (Fig 8B and 8C). Residue $E37_{UL34}$, which has

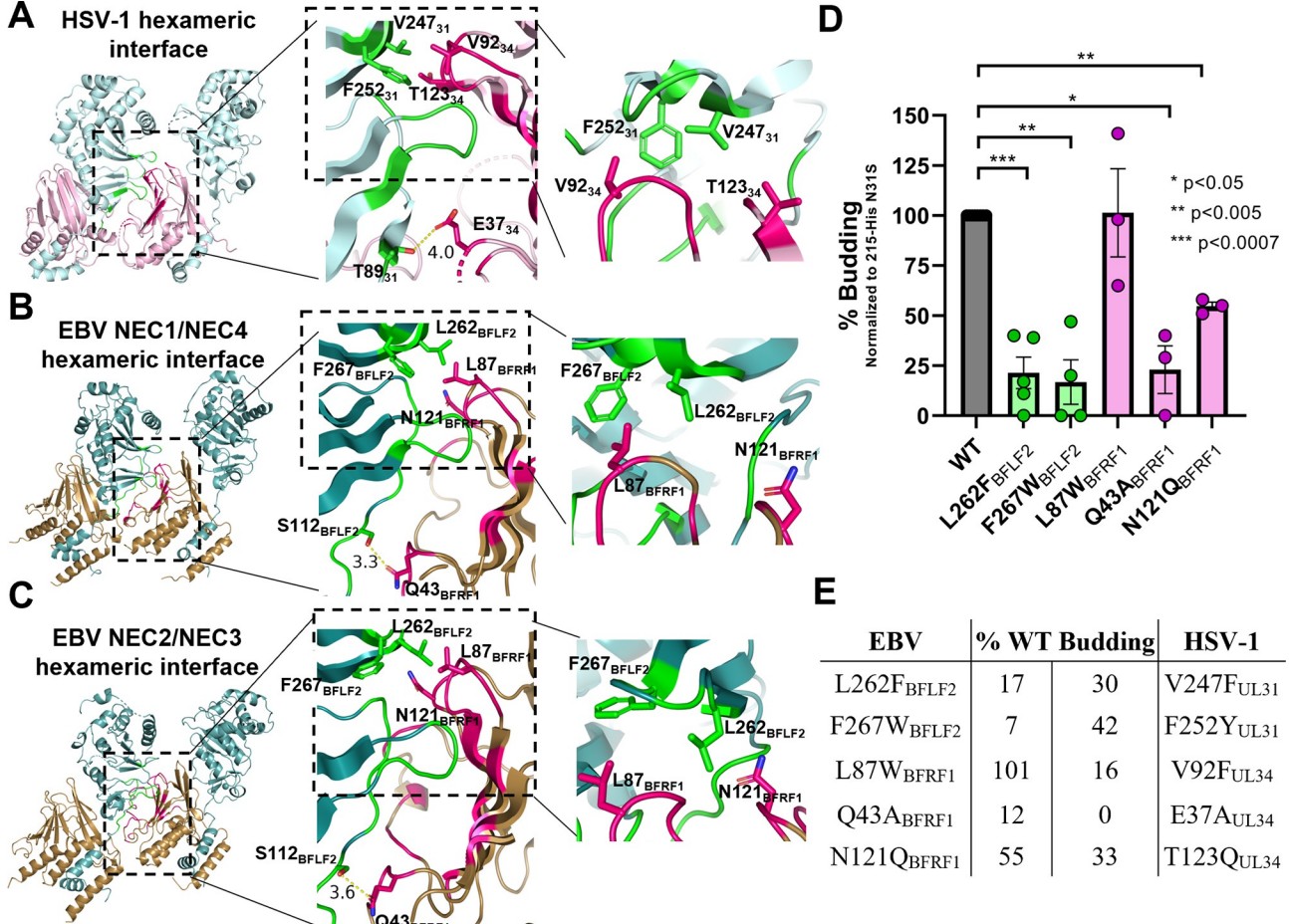

**Fig 8. Mutations to the EBV oligomeric interface reduce budding *in vitro*.** HSV-1 (A) and EBV (B and C) oligomeric interfaces with interacting residues in UL31 and BFLF2 (green) and UL34 and BFRF1 (pink). Middle inset shows a zoom of the total interaction interface. Distance measurements (Å) between EBV NEC $S112_{BFLF2}$ and $Q43_{BFRF1}$ are shown. Right inset shows a zoom of the hydrophobic triads ($V92_{UL34}$/$V247_{UL31}$/$F252_{UL31}$ in HSV-1 and $L87_{BFRF1}$/$L262_{BFLF2}$/$F267_{BFLF2}$ in EBV) and polar residue ($T123_{UL34}$ in HSV-1 and $N121_{BFRF1}$ in EBV) at the top margin of the interface. Crystal structure of the HSV-1 NEC homolog (rcsb pdb 4xzs) was used. All images created in PyMol [44]. (D) In-vitro budding assay. Vesicles contain Ni-chelating lipids to tether $His_8$-tagged NEC to membranes. % budding was determined by counting the number of ILVs after addition of NEC. Background levels of ILVs in the absence of NEC, typically around 10%, were subtracted from all values. All samples contain the $N31S_{BFLF2}$ mutation. Significance to $215$-$N31S_{BFLF2}$-$His_8$ was calculated using an unpaired Student's t-test with Welch's correction ($p < 0.05$ = *, $p < 0.005$ = **, $p < 0.0007$ = ***). In all plots, error bars represent the standard error of the mean (68% confidence interval of the mean) for at least three individual experiments. (E) Table summarizing the levels of in-vitro budding compared to wild-type for EBV mutants and their HSV-1 counterparts. Data for HSV-1 mutants obtained from [27].

the strongest non-budding phenotype in HSV-1 NEC, lacks a counterpart in EBV. Instead, we mutated a nearby residue $Q43_{BFRF1}$ that is located in an analogous position and forms a hydrogen bond with $S112_{BFLF2}$ of the neighboring NEC (Fig 8B and 8C), the equivalent of $T89_{UL31}$ (Fig 5A). All mutations were introduced into the EBV NEC215-N31S$_{BFLF2}$-His$_8$ construct, which is identical to the construct used in cryo-EM experiment except for the presence of a C-terminal His$_8$-tag in BFRF1.

We found that three out of five EBV mutations, $L262W_{BFLF2}$, $F267W_{BFLF2}$, and $Q43A_{BFRF1}$, reduced budding to <25% of the WT EBV NEC (Fig 8D and 8E). Another mutation, $N121Q_{BFRF1}$, reduced budding, ~2-fold. Notably, two out of three mutations in the EBV hydrophobic "triad", $F267W_{BFLF2}$ and $L262F_{BFLF2}$, significantly reduced *in vitro* budding, underscoring the importance of this structural element for budding in both HSV-1 and EBV. Although mutation $L87W_{BFRF1}$ did not have an apparent effect on budding, we hypothesize that position 87, located in a flexible loop, can accommodate a larger tryptophan side chain.

Therefore, we hypothesize that the dimeric interfaces observed in the EBV NEC crystals are functionally relevant and represent oligomeric interfaces formed during budding. Fig 8E compares the effectiveness of analogous interface mutations in EBV and HSV-1 NEC.

## Discussion

The NEC is essential for capsid nuclear egress across the three subfamilies of the *Herpesviridae*. But to date, most structural and mechanistic studies have focused on the NEC homologs from alpha- and betaherpesviruses leaving the NECs from gammaherpesviruses less well characterized. Here, we undertook structural and functional studies of NEC from EBV, a gammaherpesvirus. First, we demonstrated that the purified, recombinant EBV NEC vesiculates synthetic membranes *in vitro* and forms membrane-bound coats, which suggests that the intrinsic membrane budding ability is a conserved property of the NECs across *Herpesviridae*. Second, we presented the most complete crystal structure of the EBV NEC to date, which allowed us to catalog the conserved and subfamily-specific features across NEC homologs from all three subfamilies for the first time. Moreover, we presented evidence that the EBV NEC structure is intrinsically dynamic and hypothesize that such structural plasticity may be important during lattice formation. However, instead of hexamers, the EBV NEC forms dimers in the crystals, and its membrane-bound coats formed *in vitro* do not resemble the hexagonal coats formed by NEC from alphaherpesviruses [25,43]. The dimeric interfaces observed in the EBV NEC crystals are similar the hexameric interfaces observed in other NEC homologs, and mutations engineered to disrupt dimeric interfaces reduce budding. We hypothesize that membrane budding by the EBV NEC is driven by its oligomerization into membrane-bound coats but that its structural flexibility may enable formation of coats of a different geometry.

### The intrinsic membrane budding ability is a conserved NEC property

Previous studies showed that NECs from alphaherpesviruses PRV and HSV-1 budded synthetic membranes *in vitro* [25,26,50,51] in the absence of any additional factors. Although the NEC homologs from other subfamilies are expected to vesiculate the nuclear envelope around the capsid during nuclear egress, the intrinsic membrane budding ability had not been formally demonstrated until the present study. By demonstrating that the purified, recombinant EBV NEC vesiculates synthetic membranes *in vitro*, we established the intrinsic membrane budding ability as a conserved property of NECs across the *Herpesviridae* subfamilies.

The MPR of HSV-1 UL31, residues 1–50, is required for membrane interactions and budding [25,50]. Based on sequence alignments, the UL31 MPR maps to residues 1–65 in EBV BFLF2. But whereas in HSV-1, residues 1–40 of UL31 MPR are dispensable for budding, at

least, *in-vitro*, in EBV, only residues 1–15 can be removed without disrupting budding. We hypothesize that the differences in the required MPR length is due to the number and the location of dibasic motifs, which have been shown to be important for HSV-1 budding [50]. HSV-1 has five distinct basic clusters in the UL31 MPR whereas the EBV BFLF2 MPR only has one cluster (residues R22/R23) (S3B Fig). These findings suggest that the importance of the BFLF2 MPR and, possibly, basic clusters for budding is a conserved property of alpha- and gamma-herpesvirus subfamilies.

## Conserved and subfamily-specific structural features of NEC homologs

The crystal structures of the NEC homologs from the three divergent *Herpesviridae* subfamilies reveal remarkable structural similarities despite the relatively low sequence identity. The observed differences are limited to the length of the secondary structure elements, the length and the conformation of several loops, and the relative orientations of UL31 and UL34 homologs within the complex.

Importantly, the five EBV NEC heterodimers have notable structural differences, mainly, within BFLF2 and at the BFLF2/BFRF1 interface. The conformational differences among the five EBV NEC heterodimers (Fig 1E) are comparable to the differences among the NEC homologs (Fig 2E), demonstrating the remarkable structural plasticity of the EBV NEC.

## Conformational dynamics across the NEC homologs

Although the structural plasticity observed among the crystal structures of the EBV NEC is unusual among the NEC homologs, conformational dynamics may be shared by some NEC homologs to some extent. A recent molecular dynamics (MD) simulation study of the NEC crystal structures from HSV-1, PRV, and HCMV revealed differences in the intrinsic dynamics across the homologs, with HCMV NEC being more dynamic than HSV-1 or PRV NEC [53]. Whereas the "primary" hook-in-groove interface was very rigid in all three NECs homologs throughout all simulations, the globular domain of HCMV UL50 (UL31 homolog) twisted around the vertical axis to a much greater extent. These differences were attributed to the "secondary" UL31/UL34 interface.

In HSV-1 NEC, the secondary interface contains two salt bridges, $R167_{UL34}$-$D104_{UL31}$ (termed "proximal" due to its proximity to the UL31 hook) and $R158_{UL34}$-$D232_{UL31}$ (termed "distal" relative to the UL31 hook) (Fig 3C). Throughout MD simulations, both salt bridges in HSV-1 were highly stable resulting in a more rigid secondary interface and domain orientation. The distal salt bridge appears to be important for function because the double mutation $R158A_{UL34}$/$R161A_{UL34}$ reduced viral titers, suggesting that these charged residues are involved in NEC function [54]. The mutation does not interfere with NEC localization [54], nor does it result in a dominant-negative phenotype [31], implying that it does not block NEC/NEC interactions. By contrast, in HCMV, there is a single, proximal salt bridge $E147_{UL50}$-$R102_{UL53}$ and a distal hydrogen bond between $N139_{UL50}$ and $Q103_{UL53}$ instead of a distal salt bridge (Fig 3C). Both the salt bridge and the hydrogen bond in HCMV were less stable than the two salt bridges in HSV-1 NEC, suggesting that the more flexible HCMV domain orientation is due to a less stable secondary interface. Importantly, when a second, distal salt bridge was introduced computationally, the HCMV NEC structure became much less dynamic in the MD simulations, which suggests that two salt bridges increase the conformational stability. While the HSV-1 and HCMV NEC heterodimers differed in domain orientation flexibility on their own, in the context of a hexagonal arrangement the domain twisting was drastically reduced [53]. The authors speculated that the increased HCMV flexibility may influence events prior to nuclear egress and hexamer formation such as interactions with NEC-associated proteins.

The EBV NEC structure is more similar to that of HCMV because it has a proximal salt bridge, $D115_{BFRF1}$-$R128_{BFLF2}$, in four heterodimers (except the incomplete NEC5) but no distal salt bridges (Fig 3B). NEC1 and NEC4 also have a distal hydrogen bond between $T158_{BFRF1}$ and $E250_{BFLF2}$. The structural dynamics of the EBV NEC, observed experimentally, correlates with the structural dynamics of the HCMV NEC heterodimer revealed in the MD simulations. However, because these structural dynamics are greatly reduced once HCMV NEC is arranged into a hexagon, we hypothesize that the conformational plasticity of EBV NEC may influence its oligomerization into a coat. Further studies are required to understand the influence of EBV NEC structural flexibility on coat formation.

## The ability of the NEC to oligomerize is conserved

The ability of the EBV NEC to oligomerize is supported by our cryo-EM images, the EBV NEC dimers observed in the crystals, and by the similarities between the dimeric interfaces observed in the EBV NEC crystals and the hexameric interfaces observed in HSV-1 NEC [27] and, to a lesser extent, HCMV NEC [39]. Moreover, mutations engineered to disrupt the dimeric interface reduce budding to the extent similar to interface mutations in HSV-1 NEC [27]. Comparisons revealed conserved structural elements across these oligomeric interfaces, notably, a hydrophobic triad at the membrane-distal margin of the interface. Mutations designed to increase the size of the hydrophobic side chain in the triad so as to disrupt oligomerization reduced HSV-1 budding *in vitro* ($F252Y_{UL31}$, $V247F_{UL31}$, and $V92F_{UL34}$) [27], and in infected cells ($V247F_{UL31}$) [32]. We found that two out of three corresponding mutations in EBV NEC, $F267W_{BFLF2}$ and $L262F_{BFLF2}$, significantly reduced *in vitro* budding, stressing the importance of the hydrophobic triad at the oligomeric interface for budding in both HSV-1 and EBV. We propose that the dimeric interfaces observed in the EBV NEC crystals are functionally relevant and represent interfaces formed during budding.

Previous work established that the NECs from alphaherpesviruses PRV and HSV-1 oligomerize into hexagonal coats on the inner surface of budded vesicles formed *in vitro* (HSV-1) [25,43], in uninfected cells overexpressing the NEC (PRV) [29], and in perinuclear enveloped vesicles purified from infected cells (HSV-1) [30]. Despite the predominance of hexagonal coats, both NEC pentamers [43] and heptamers [30] have also been observed for HSV-1 under certain conditions suggesting HSV-1 NEC can arrange into alternative oligomeric assemblies. Given that a purely hexagonal arrangement cannot yield a closed sphere, it was proposed that incorporation of pentamers or hexamers into a hexameric NEC coat could generate curvature. A similar strategy is utilized by the poxvirus protein, D13L, which forms a curved lattice composed of mainly hexamers, incorporated with pentamers and heptamers [55]. It remains unknown exactly how HSV-1 NEC curvature is achieved.

Given structural similarities between the NEC homologs, we anticipated that EBV NEC would form hexamers in the crystals and hexagonal membrane-bound coats during budding *in-vitro*. Instead, we found that EBV NEC formed dimers in the crystals and that its membrane-bound coats appeared different than the HSV-1 NEC coats. We hypothesize that the high degree of structural plasticity observed among the five EBV NEC heterodimers in the crystals could potentially influence NEC oligomerization and coat geometry. One possibility is that the EBV NEC plasticity results in the formation a pseudo-hexagonal lattice containing multiple irregular defects, reminiscent of the immature hexameric Gag lattice of HIV capsids [56,57]. As hypothesized for HSV-1 NEC, such irregular defects within the EBV NEC coat could serve to accommodate curvature around the capsid. An alternative possibility is that the EBV NEC coats have an alternative, previously unobserved NEC arrangement, e.g., stacks of NEC dimers.

We propose that membrane budding by the EBV NEC is driven by its oligomerization into membrane-bound coats but that its structural flexibility enables formation of coats of a different geometry. Future high-resolution cryo-ET reconstructions will uncover the precise structural arrangement of EBV NEC within the membrane-bound coats.

## Materials and methods

### Plasmid cloning

Codon-optimized EBV (strain B95-8) BFRF1 gene was synthesized by Life Technologies and used as a template to produce PCR fragments encoding BFRF1 1–195 (pJB77 in S8 Table) and BFRF1 1–228 (pJB73). All primers are listed in S8 Table. The PCR products were purified, digested, and ligated into pGEX-6P-1 to produce constructs containing an N-terminal glutathione S-transferase (GST) tag followed by a PreScission cleavage site, for increased solubility and affinity purification. BFRF1 1-228-His$_8$ (pMT47), with a C-terminal His$_8$-tag for tethering to Ni-chelating lipids, was generated by PCR and subcloned into pGEX-6P-1 containing a C-terminal His$_8$-tag. EBV BFRF1 1-215-His$_8$ (pMT46) was generated by inverse PCR followed by blunt end ligation. EBV BFRF1 1-215-His$_8$ mutants L87W (pMT54), Q43A (pMT56), and N121Q (pMT57) were generated by site-directed mutagenesis using overlapping inverse PCR followed by DpnI digestion and transformation into *Escherichia coli* DH5α cells.

A gene for EBV (strain B95-8) BFLF2 (a gift from Dr. Prashant Desai) used as a template to produce PCR fragments encoding EBV BFLF2 1–318 (pMT05) and EBV BFLF2 66–318 (pJB75). The PCR products were purified, digested, and ligated into pET24b with an N-terminal His$_{10}$-SUMO tag followed by a PreScission cleavage site, for increased solubility and affinity purification. N-terminally truncated EBV BFLF2 16–318 (pMT37), BFLF2 25–318 (pMT38), BFLF2 35–318 (pMT39), and BFLF2 45–318 (pMT40) were generated using inverse PCR followed by blunt end ligation. Due to a cleavage event at N31, the N31S (pMT49) mutation was introduced into EBV BFLF2 1–318 by inverse PCR followed by blunt-end ligation. EBV BFLF2 1–318 mutants N31S/L262F (pMT59) and N31S/F267W (pMT62) were generated by site-directed mutagenesis using overlapping inverse PCR followed by DpnI digestion and transformation into *Escherichia coli* DH5α cells.

### Protein expression and purification

Plasmids encoding EBV BFRF1 and BFLF2 constructs were co-transformed into *E. coli* Rosetta (DE3) cells and expressed in TB medium at 25˚C for 16 hours following lactose-derived auto-induction [58]. The EBV NEC complexes were named according to the previously used nomenclature for HSV-1 NEC complexes [25]. For example, EBV NEC195Δ65 is composed of BFRF1 1–195 and BFLF2 66–318. The purification process has been described previously [50]. Briefly, cells were harvested by centrifugation, resuspended in lysis buffer (50 mM buffer see S9 Table, 500 mM NaCl, 0.5 mM TCEP, 10% glycerol) in the presence of Complete protease inhibitor (Sigma-Aldrich) and DNase (Sigma), and lysed using a M-110S microfluidizer (Microfluidics). Cell lysate was centrifuged at 17,370 x g in a Beckman J2-21 floor centrifuge. All purification steps were performed at 4˚C. Clarified cell lysate was loaded onto the Ni-NTA sepharose resin (GE Healthcare). Resin was washed with wash buffer (lysis buffer plus 20–40 mM imidazole). Bound protein was eluted with elution buffer (lysis buffer plus 250 mM imidazole) and loaded onto glutathione Sepharose resin (GE Healthcare) to remove excess His$_6$-SUMO-UL31. Resin was washed with wash buffer (lysis buffer plus 1 mM EDTA). Removal of the His$_6$-SUMO and GST tags by cleavage with a GST-PreScission protease (produced in-house) was performed on the glutathione sepharose column for 16 hours. Lysis buffer was used to elute the His$_6$-SUMO and cleaved NEC from the glutathione sepharose column. EBV

NEC228Δ15-His$_8$, NEC228Δ24-His$_8$, NEC215-Q43A$_{BFRF1}$/N31S$_{BFLF2}$-His$_8$, and NEC215-N121Q$_{BFRF1}$/N31S$_{BFLF2}$-His$_8$ were then passed over a HisTrap HP column (GE Healthcare) and fractions containing NEC were concentrated to 500 μL. EBV NEC195Δ65, NEC228Δ44-His$_8$, NEC228Δ34-His$_8$, NEC215-N31S$_{BFLF2}$-His$_8$, NEC215-L87W$_{BFRF1}$/ N31S$_{BFLF2}$-His$_8$, NEC215-N31S$_{BFLF2}$/L262F$_{BFLF2}$-His$_8$, NEC215-N31S$_{BFLF2}$/F267W$_{BFLF2}$-His$_8$ constructs were not passed over the HisTrap HP column and instead were eluted from the glutathione Sepharose resin and concentrated to 500 μL. Once concentrated to 500 μL, all EBV NEC constructs were passed over an S75 10/300 size-exclusion column (GE healthcare) with the gel-filtration buffer (20 mM buffer see S9 Table, 100 mM NaCl, 0.5 mM TCEP) to separate NEC from His$_6$-SUMO. EBV NEC215-N31S$_{BFLF2}$ was diluted to 100 mM NaCl after elution from the glutathione Sepharose resin with 50 mM MES pH 6.0, 0.5 mM TCEP, 10% glycerol. EBV NEC215-N31S$_{BFLF2}$ was separated from His$_6$-SUMO using a cation exchange resin (HiTrap SP XL, GE Healthcare) with a 100 mM to 600 mM NaCl gradient in 20 mM MES pH 6.0, 0.5 mM TCEP. Each EBV NEC construct was purified to homogeneity as assessed by 12% SDS-PAGE and Coomassie staining.

Fractions containing EBV NEC were concentrated to ~5 mg/mL using Amicon Ultra-4 50 kDa cutoff centrifugal filter units (EMD Millipore) and stored at -80˚C to avoid aggregation and degradation observed during storage at 4˚C. Protein concentration was determined by absorbance measurements at 280 nm using theoretical extinction coefficient of 17,880 calculated using ProtParam (https://web.expasy.org/protparam/). The typical yield was ~1.5 mg per L of TB culture. Due to different isoelectric points for numerous EBV NEC constructs, pH of the buffers was adjusted accordingly (see S9 Table). Lysis buffer always contained 50 mM buffer, 500 mM NaCl, 0.5 mM TCEP, 10% glycerol and gel-filtration buffer always contained the final concentration of 20 mM buffer, 100 mM NaCl and 0.5 mM TCEP.

HSV-1 NEC220 was expressed and purified with the same techniques as EBV NEC215-N31S$_{BFLF2}$. The buffers can be found in S9 Table.

## Crystallization and data collection

Crystals of native EBV NEC195Δ65 were grown by vapor diffusion at 20˚C in hanging drops containing 2 μL protein in gel filtration buffer (20 mM HEPES pH 8.0, 100 mM NaCl, 0.5 mM TCEP) equilibrated over 500 μL reservoir solution (25% PEG 3350, 0.1 M Tris-HCl pH 8.5, 0.2 M Li$_2$SO$_4$). No reservoir solution was added to the drops. The crystals were identified using an automated microscope [59,60]. Crystals would not grow in a 1:1 hanging drop of protein:reservoir solution. Tetragonal crystals appeared after 1 day and grew to their final size after 2 days. Crystals were cryopreserved using ethylene glycol in a step-wise cryoprotectant transfer protocol [61]. Briefly, crystals were harvested with a loop and transferred into a 2 μL drop containing the harvesting buffer (1.5x gel filtration buffer and no ethylene glycol). Then, 2 μL of the harvesting buffer plus 5% ethylene glycol was added on one side and then 2 μL was removed from the other side, taking care to leave the crystal in the drop. This process was repeated sequentially with the harvesting buffer plus 10%, 15%, 20%, 25%, 30%, 35%, 40% ethylene glycol. The crystal was then flash frozen in liquid nitrogen.

Initially, native diffraction data were collected at the Advanced Photon Source at Argonne National Labs at beamline 24ID-E at the wavelength of 0.979180 Å at 100 K and processed to 3.93 Å using XDS [62] as implemented in the Northeastern Collaborative Access Team (NE-CAT) rapid automated processing of data (RAPD) software (https://rapd.nec.aps.anl. gov). Crystals took the tetragonal space group P4$_{1/3}$2$_1$2 with unit cell dimensions a = b = 238.698 Å, c = 138.074 Å and α = β = γ = 90˚. Molecular replacement as implemented in Phaser [63] using the crystal structures of NEC homologs from HSV-1, PRV, or HCMV (RCSB

PDB 4zxs, 4z3u, or 5d5n, respectively) did not yield any solution. Only the structure of the EBV NEC "base" (RCBS PDB 6t3z), composed of EBV BFRF1 1–192 bound to the "hook" fragment of BFLF2 59–87, yielded a correct solution for four copies of the EBV NEC base in space group $P4_32_12$ (LLG = 929). However, no connected density was observed for the globular domain of BFLF2.

All known NEC structures contain a Zn ion coordinated by four strictly conserved residues within the UL31 homologs, three cysteines and one histidine. Therefore, we hypothesized that the EBV NEC also contained a bound Zn that could be used for structure determination by single-wavelength anomalous dispersion (SAD). Fluorescence scans on an EBV NEC crystal at the APS beamline 24ID-C confirmed the presence of Zn. SAD data were collected at the wavelength of 1.28215 Å (Zn) at 100 K, and two data sets from different crystals were merged using RAPD and processed together to 3.97 Å. Four high-occupancy Zn sites were located using the SHELX macromolecular substructure solution program [64], and the Phenix AutoSol program [65] was used to calculate experimental phases and generate a preliminary electron density map, within the RAPD SAD pipeline. However, the electron density did not permit either autotracing in Autosol or manual tracing.

Therefore, the structure of the EBV NEC was ultimately solved by a combination of molecular replacement and Zn SAD. First, phases calculated from the molecular replacement solution obtained using the structure of the EBV NEC base (RCBS PDB 6t3z) were used to locate the four Zn sites in the SAD dataset, in Autosol [65]. Each Zn site was located near a base where it would be predicted to be based on the expected BFLF2 location. A density modified map revealed the density for the three copies of BFLF2. Due to the poor density, autotracing was not feasible. Therefore, the structure of the globular core of HCMV UL53 (a homolog of EBV BFLF2 and HSV1- UL31), from RSCB PDB 5d5n, was manually placed into the density modified map and underwent extensive rebuilding in Coot [66]. The initial model was refined against the SAD data in phenix.refine [65] to 3.97 Å. Prior to refinement, 5% of all data were set aside for cross-validation. After several cycles of refinement and model rebuilding, the density for the 4th copy of BFLF2 became interpretable, which allowed its manual placement. Surprisingly, after additional rounds of refinement and rebuilding, the density for the 5th copy of EBV NEC became evident, and the 5th copy was built into it.

Model refinement included gradient minimization refinement of xyz coordinates and individual thermal parameters with optimization of X-ray/stereochemistry and X-ray/ADP weights, and secondary structure restraints. Rigid-body refinement was employed during early stages of refinement. Experimental phase restraints were used until the late stages of refinement. Electron density maps were sharpened. The final model encompasses five polypeptide chains of BFRF1 (A, C, E, G, and I), five polypeptide chains of BFLF2 (B, D, F, H, and J), and five Zn ions. Side chains were modeled into the available electron density. When electron density for the side chains was missing, the most common rotamer was used. According to MolProbity [67] as implemented in phenix.refine, 92.61% of residues lie in the most favored regions of the Ramachandran plot and 6.29% lie in the additionally allowed regions. The overall fitting of the model to the electron density yielded an R-free of 0.308. The following residues are resolved in each chain: Chain A, 1–194 (99%), Chain B, 78–316 (89%), Chain C, 1–194 (99%), Chain D, 77–318 (91%), Chain E, 1–194 (99%), Chain F 78–150, 157–160, 163–171, 175–254, 258–271, 283–292, 299–316 (77%), Chain G, 1–194 (99%), Chain H, 78–318 (91%), Chain I, 4–20, 42–48, 51–75, 80–82, 85–93, 96–110, 113–126, 129–191 (78%), Chain J, 83–135, 137–153, 160–171, 174–185, 192–221, 223–279, 295–316 (75%). Relevant crystallographic statistics are listed in Table 1.

## Liposome preparation

Liposomes were prepared as described previously [25]. Briefly, lipids were mixed in a molar ratio of 58% POPC/11% POPE/9% POPA/9% POPS/5% cholesterol/5% DGS-NTA/3% POPE Atto594. 5 μL was then spread on the surface of an ITO-covered slide and vacuum desiccated for 30 minutes. Once dry, a vacuum-greased O-ring was placed around the lipid mixture and the VesiclePrep Pro (Nanion Technologies) was used to produce an AC field (sinusoidal wave function with a frequency of 8Hz and amplitude 2V) before adding 270 μL of lipid swelling buffer (300mM sucrose dissolved in 5 mM Na-HEPES, pH 7.5). A second ITO-covered slide was then placed to cover the lipid/buffer mixture after 3 minutes followed by a 2-hour swell and a 5-minute fall step. GUVs were used immediately and diluted 1/20 with 20 mM Tris pH 8.5, 100 mM NaCl, 0.5 mM TCEP.

## GUV budding assay

Fluorescently labeled GUVs were co-incubated with the soluble NEC and the membrane impermeable dye, Cascade Blue Hydrazide (ThermoFisher Scientific) for 3 minutes. The GUVs contained 18% negatively charged lipids (58% POPC/11% POPE/9% POPA/9% POPS/ 5% cholesterol/5% DGS-NTA/3% POPE Atto594), which resembles the INM of uninfected cells [68,69]. The nuclear membrane composition during HSV-1 infection is unknown but may differ from that of uninfected cells. Budding events display as intraluminal vesicles (ILVs) containing Cascade Blue within the GUVs (Fig 6C). 5 μL of the above GUV composition and 2 μL Cascade Blue Hydrazide were mixed with a final concentration of 1.5 μM NEC for a total volume of 100 μL. Each sample was visualized using a Leica SP8 confocal microscope. Background levels of ILVs, counted in the absence of NEC, were subtracted from counts of ILVs in the presence of NEC. Background levels are typically around 10%. Experiments were performed with at least 3 technical replicates and at least 3 biological replicates. The standard error of the mean is reported from at least three individual experiments. Data was plotted using GraphPad Prism 9.0.

## Cryo-electron microscopy

Purified HSV-1 NEC220 (1 mg/mL; 30 μL) or EBV NEC215-N31S$_{BFLF2}$ (0.9 mg/mL; 35 μL) were incubated with 10 μL of a 1:1 mixture of 400:800 nm large unilamellar vesicles (LUVs) (made of 3:1:1 POPC:POPS:POPA as previously described in [25]) on ice for 30 min. At the end of incubation, 3 μL of each sample were applied to Lacey Carbon copper grids (50 nm hole size, 300 mesh, Electron Microscopy Sciences) and blotted on both sides for 6 s. The grids were rapidly vitrified in liquid ethane (Vitrobot) and stored in liquid nitrogen until loaded into a Tecnai F30 transmission electron microscope (FEI) using a cryo holder (Gatan). The microscope was operated in low-dose mode at 300 keV using the SerialEM software [70]. Images were recorded with a Ultrascan 2k x 2k post-filter GIF camera at either 31,000-fold (pixel size: 0.362 nm) or 39,000-fold magnification (pixel size: 0.287 nm) at defocus values of -4 to -7 um and an electron dose of ~7–10 e/Å$^2$. Images are displayed using Fiji [71].

## Supporting information

**S1 Fig. Secondary structure assignments and structural alignments of BFRF1, BFLF2, and their homologs.** (A) Crystal structures of EBV BFRF1 (rcsb pdb 7t7i) and homologs from HCMV UL50 (rcsb pdb 5d5n), HSV-1 UL34 (rcsb pdb 4xzs), and PRV UL34 (rcsb pdb 4z3u). Secondary structure assignments were obtained from DSSP [73]. Secondary structures are color coded, α-helices (light pink) and β-strands (dark pink). (B) Crystal structures of EBV

BFLF2 (rcsb pdb 7t7i) and homologs from HCMV UL53 (rcsb pdb 5d5n), HSV-1 UL31 (rcsb pdb 4xzs), and PRV UL31 (rcsb pdb 4z3u). Secondary structure assignments were obtained from DSSP [73]. Secondary structures are color coded, α-helices (light blue) and β-strands (dark blue). All images created in PyMol [44].
(TIF)

**S2 Fig. Sequence alignment of EBV BFRF1 and BFLF1.** Alignment of sequences for BFRF1 (A) and BFLF2 (B). Secondary structure from the crystal structure is shown above sequence alignment blocks. Vertical black arrows denote EBV crystal construct boundaries. Unresolved residues are shown in grey boxes. Residues at the arm-hook interaction are colored, BFRF1 (cyan) and BFLF2 (red). Residues at the globular interface are colored, BFRF1 (green) and BFLF2 (yellow). Alignment made with Clustal Omega [47] and annotated using Esprit [48]. Interface analysis done with PDBePISA [46].
(TIF)

**S3 Fig. Sequence alignment of EBV and homologous NECs.** Alignment of sequences for BFRF1 and homologs (A) and BFLF2 and homologs (B). Secondary structures from crystal structures are shown above sequence alignment blocks. Vertical black arrows denote EBV crystal construct boundaries. Alignment made with Clustal Omega [47] and annotated using Esprit [48]. Interface analysis done with PDBePISA [46]. Residues at the arm-hook interaction are colored, BFRF1 homologs (cyan) and BFLF2 homologs (red). Residues at the globular interface are colored, BFRF1 homologs (green) and BFLF2 homologs (yellow). Secondary structures from HCMV (rcsb pdb 5d5n), HSV-1 (rcsb pdb 4xzs), and PRV (rcsb pdb 4z3u).
(TIF)

**S1 Table. Resolved residues for each individual EBV BFRF1 (top) and BFLF2 (bottom) crystal structure chains.** For each EBV chain, the boundaries of resolved residues and % resolved residues are listed.
(DOCX)

**S2 Table. Structural alignments of the five EBV NEC heterodimers in the asymmetric unit.** Heterodimers (NEC) and individual BFRF1 or BFLF2 chains were aligned. For each EBV NEC or chain, RMSD (Å) is listed followed by the number of aligned residues in parentheses. In all cases, "SSM superpose" in WinCoot [66] was used to carry out the structure alignments and calculate RMSDs, except for those denoted by * in which case "LSQ Superpose" command was used.
(DOCX)

**S3 Table. Sequence conservation among homologous herpesvirus NEC components.** Clustal Omega [47] was used for sequence alignment and calculations of % identity. The following UniProtKB IDs were used: HSV-1 UL34 (P10218), HSV-1 UL31 (P10215), PRV UL34 (G3G8R3), PRV UL31 (G3G955), HCMV UL50 (P16791), HCMV UL53 (P16794), EBV BFRF1 (P03185), and EBV BFLF2 (P0CK47).
(DOCX)

**S4 Table. Structural alignments of the EBV NEC to its homologs in HSV-1, PRV, and HCMV.** For each EBV NEC or chain, RMSD (Å) is listed followed by the number of residues aligned in parentheses. In all cases, "SSM Superpose" in WinCoot [66] was used to carry out the structure alignments and calculate RMSDs, except for those denoted by * in which case "LSQ Superpose" command was used. RMSD and residues aligned comparisons for BFLF2 chains B-J are not available for EBV due to the presence of only 32 residues of BFLF2 in rcsb pdb 6t3z [42]. Crystal structures of EBV BFRF1 (rcsb pdb 6t3z) and NEC homologs from

HSV-1 (rcsb pdb 4xzs), PRV (rcsb pdb 4z3u and 5e8c), and HCMV (rcsb pdb 5d5n and 5dob) were used.
(DOCX)

**S5 Table. Interface areas for the globular, hook-in-groove, and total heterodimeric interfaces for the EBV BFLF2:BFRF1 interaction.** PDBePISA [46] analysis of the interface area between the globular domains, hook-in-groove and total heterodimeric interface. The globular interface area was obtained by deleting the arm-hook portion of BFLF2/HSV-1 UL31/PRV UL31/HCMV UL53 from files before analysis ending at V111/K88/K55/G88, respectively. The arm-hook interaction was obtained by subtracting the globular interface area from the total heterodimeric interaction. Total heterodimeric interface area was obtained by inputting the entire crystal structure. Crystal structures the NECs homologs from HSV-1 (rcsb pdb 4xzs), PRV (rcsb pdb 4z3u and 5e8c), and HCMV (rcsb pdb 5d5n and 5dob) were used.
(DOCX)

**S6 Table. Interface areas for the oligomeric interfaces of EBV NEC and all applicable homologous NEC crystal structures.** PDBePISA [46] analysis of the solvent accessible area buried at the oligomeric interface. Crystal structures of the NEC homologs from HSV-1 (rcsb pdb 4xzs) and HCMV (rcsb pdb 5d5n) were used.
(DOCX)

**S7 Table. Conservation of residues at the oligomeric interfaces.** Interface residues in the two EBV NEC dimers in the crystal structure were mapped using PDBePISA [46] analysis (Fig 4). Interface residues that are identical across in BFLF2 and its homologs or BFRF1 and its homologs from EBV, HSV-1, and HCMV (Identical interface residues) were divided by the total number of interface residues in the respective dimer (Total interface residues) to yield % Identical Interface Residues. % Total Sequence Conservation is the % identical residues in BFLF2 or BFRF1.
(DOCX)

**S8 Table. List of primers used for cloning described in Materials and Methods.** Mutations are bolded, restriction digest sites are underlined and listed underneath applicable primers. BFRF1 constructs are codon optimized (c.o.).
(DOCX)

**S9 Table. Protein purification buffers.** Due to different isoelectric points (PIs) of EBV NEC constructs, pH of buffers used for purification was adjusted to be ~1 pH unit away from the pI. Cells were lysed in lysis buffer (50 mM indicated buffer, 500 mM NaCl, 0.5 mM TCEP, 10% glycerol). Affinity tag removal changed the PIs, so buffer with a different pH was used during and after PreScission protease cleavage. After affinity tag removal, protein was further purified in gel filtration buffer (20 mM indicated buffer, 100 mM NaCl, 0.5 mM TCEP). Salt concentration varied for ion exchange chromatography (see Materials and Methods).
(DOCX)

## Acknowledgments

We thank Janna Bigalke for generating plasmids pJB73, pJB75, and pJB77 and testing protein expression; Isabelle Mueller for help with the budding experiments on the N-terminal truncation mutants of BFLF2; Gonzalo Gonzalez-Del Pino for help with x-ray data processing and structure refinement; and Surajit Banerjee at APS for his help and expertise in x-ray data collection and processing. We thank Rob Jackson (Tufts University School of Medicine) and

Martin Hunter (University of Massachusetts College of Engineering) for assistance with fluorescence microscopy experiments. We thank Andrew Bohm (Tufts University School of Medicine) for use of his automated microscope for protein crystal screening. We also thank the staff at the NE-CAT (Advanced Photon Source) for help with collecting X-ray diffraction data, Peter Cherepanov (Francis Crick Institute) for the gift of the GST-PreScission protease expression plasmid, and Thomas Schwartz (Massachusetts Institute of Technology) for the gift of LoBSTr cells. Cryo-EM samples were prepared and imaged at the Brandeis Electron Microscopy Facility. Confocal microscopy was performed at the Tufts Center for Neuroscience Research at Tufts University School of Medicine supported by NIH grant P30 NS047243 (Rob Jackson). This work is based upon research conducted at the Northeastern Collaborative Access Team beamlines, which are funded by the National Institute of General Medical Sciences from the National Institutes of Health (P41 GM103403). The Pilatus 6M detector on 24-ID-C beam line is funded by a NIH-ORIP HEI grant (S10 RR029205). This research used resources of the Advanced Photon Source; a U.S. Department of Energy (DOE) Office of Science User Facility operated for the DOE Office of Science by Argonne National Laboratory under Contract No. DE-AC02-06CH11357. All software was installed and maintained by SBGrid [72].

## Author Contributions

**Conceptualization:** Michael K. Thorsen, Ekaterina E. Heldwein.

**Data curation:** Michael K. Thorsen, Ekaterina E. Heldwein.

**Formal analysis:** Michael K. Thorsen, Elizabeth B. Draganova, Ekaterina E. Heldwein.

**Funding acquisition:** Ekaterina E. Heldwein.

**Investigation:** Michael K. Thorsen, Elizabeth B. Draganova.

**Methodology:** Michael K. Thorsen, Elizabeth B. Draganova, Ekaterina E. Heldwein.

**Project administration:** Ekaterina E. Heldwein.

**Resources:** Ekaterina E. Heldwein.

**Supervision:** Ekaterina E. Heldwein.

**Validation:** Michael K. Thorsen.

**Visualization:** Michael K. Thorsen, Elizabeth B. Draganova.

**Writing – original draft:** Michael K. Thorsen, Ekaterina E. Heldwein.

**Writing – review & editing:** Michael K. Thorsen, Elizabeth B. Draganova, Ekaterina E. Heldwein.

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
