## [Decision Letter · Decision Letter 0]

7 Feb 2022

Dear Dr. Heldwein,

Thank you very much for submitting your manuscript "The nuclear egress complex of herpesviruses buds membranes through a conserved oligomerization-driven mechanism." for consideration at PLOS Pathogens. As with all papers reviewed by the journal, your manuscript was reviewed by members of the editorial board and by several independent reviewers. In light of the reviews (below this email), we would like to invite the resubmission of a significantly-revised version that takes into account the reviewers' comments.

As you will see in the comments that have been assembled by three highly qualified reviewers who have looked at this topic from distinct technical positions, there is an enormous range of views of the submitted manuscript.

To me, this means that you must carefully review any comments regarding data that have been raised by the more critical reviewers, but also be sure to describe prior studies clearly in order to assemble a text that makes the story clearly understood by all. Thus, the revision is in your hands, but it is important to emphasize that this will be reevaluated and editors will be sensitive to seeing better agreement amongst reviewers.

We cannot make any decision about publication until we have seen the revised manuscript and your response to the reviewers' comments. Your revised manuscript is also likely to be sent to reviewers for further evaluation.

Sincerely,

Edward S. Mocarski

Associate Editor

PLOS Pathogens

Urs Greber

Section Editor

PLOS Pathogens

Kasturi Haldar

Editor-in-Chief

PLOS Pathogens

orcid.org/0000-0001-5065-158X

Michael Malim

Editor-in-Chief

PLOS Pathogens

orcid.org/0000-0002-7699-2064

As you will see in the comments that have been assembled by three highly qualified reviewers who have looked at this topic from distinct technical positions, there is an enormous range of views of the submitted manuscript.

To me, this means that you must carefully review any comments regarding data that have been raised by the more critical reviewers, but also be sure to describe prior studies clearly in order to assemble a text that makes the story clearly understood by all. Thus, the revision is in your hands, but it is important to emphasize that this will be reevaluated and editors will be sensitive to seeing better agreement amongst reviewers.

Reviewer's Responses to Questions

**Part I - Summary**

Reviewer #1: Thorsen and Heldwein have determined the structure of BFRF1 in complex with BFLF2 - the nuclear egress complex (NEC) from EBV. Despite considerable variation in amino acid sequences between EBV NEC components and NEC proteins from HSV-1, PRV and HCMV the structure of the EBV NEC appears very similar to the alpha- and betaherpesvirus NECs suggesting the gammaherpesviruses utilize a similar mechanism for primary envelopment of capsids at the inner nuclear membrane. While a hexagonal lattice of NEC components was observed in cryo-ET reconstructions of the HSV-1 NEC as well as in the NEC crystals formed by HSV-1 and HCMV NEC components, no hexagonal lattice was observed in the EBV NEC crystals, nor was it investigated in in vitro membrane budding assays as done in the past by this group for the HSV-1 NEC. Despite these apparent differences, NEC/NEC interaction interfaces were identified in the EBV NEC crystals that resembled those formed between NEC/NEC heterodimer pairs in HSV-1 and HCMV NEC hexamers. Mutation of amino acids residues within the putative EBV NEC/NEC interface prevented membrane budding in an in vitro budding assay. The authors’ central conclusions are that the EBV NEC oligomerizes and that the putative oligomerization interface identified is required for budding. The authors further suggest that this study supports the idea that NECs from three subfamilies of the Herpesviridae operate through a common mechanism driven by the formation of a hexagonal protein lattice. Unfortunately, the data presented fail to support these conclusions. While it is appreciated that the work described is important and needed to be done to "close the loop" on NEC structural determinations of alpha-, beta- and gammaherpesviruses, the findings do not move our understanding of NEC form, or function, significantly forward.

Reviewer #2: Thorsen and Heldwein explore the structure and function of the Epstein-Barr virus (EBV) nuclear egress complex (NEC) and present data supporting the hypothesis that the NECs of herpesviruses are constitutive budding machines and that they mediate budding by a conserved mechanism requiring formation of hexamers. The simple addition of another structure to the collection of herpesvirus NEC structures might not, in itself, meet the standards for originality of this journal. However, the results of the study are very interesting primarily because of unexpected differences between the structures (plural) of the EBV NEC compared to those of other herpesviruses of the alpha and beta sub-families. Unlike its homologs, the EBV NEC is structurally heterogeneous in crystals and does not form hexamers. The structural data supporting this conclusion are compelling and suggest that the EBV NEC is more flexible than its counterparts. The authors also present strong evidence showing that the EBV NEC, like its homologs, can induce budding on synthetic membranes in vitro. They present somewhat weaker evidence that this budding is sensitive to mutations in residues homologous to those that support hexamer formation in other herpesviruses.

The paper is well written and appropriately referenced.

The data in the paper are quite straightforward and there is little to argue with there. The authors are also appropriately cautious with conclusions. The data supporting the authors proposal that the EBV NEC mediates budding by hexamer formation are weak, based on the in vitro behavior of a few mutations, not all of which have the expected effect. Possible reasons for the unexpected budding activity of the L87W mutation should be addressed in the discussion. A previous publication by these authors has suggested that membrane curvature in budding results from lipid ordering induced by the aggregated membrane proximal region(s) of the NEC. Might these results be consistent with the idea that the NEC need not specifically form hexamers, but that other aggregation arrangements might serve the purpose? This might be an entirely wrong-headed suggestion, but it would be interesting if the authors provided some speculation as to how the structural flexibility of the EBV NEC might fit into their ideas about how membrane curvature is induced.

Reviewer #3: In this manuscript, Thorsen and Heldwein examine the structure and assembly of the nuclear egress complex (NEC) of Epstein-Barr Virus, which is a member of the gamma-herpesvirus family. Prior work from the Heldwein and other groups have examined the NECS from other herpesviruses from the alpha- and beta-herpesviruses, but our understanding of the structure and function of these complexes in the gamma-herpesviruses has remained incomplete. While a previous manuscript has reported the structure of a fragment of the EBV NEC, here the authors determine a more complete structure of the BFLF2/BFRF1 heterodimer, albeit at moderate resolution, providing evidence that the EBV NEC oligomerizes through similar interactions to other NECS. In addition, the authors demonstrate that mutations in the N-terminal region and at the NEC oligomerization interface interfere with NEC-mediated budding, establishing that the herpesviruses utilize a common mechanism for inducing membrane budding to traverse the nuclear membrane. This manuscript contributes to our global understanding of herpesvirus assembly mechanisms through these structural and functional studies.

**Part II – Major Issues: Key Experiments Required for Acceptance**

Reviewer #1: Major concerns:

1) Major shortcomings include the failure to demonstrate: 1) NEC oligomerization outside of a crystal; 2) that the mutations introduced into the putative oligomerization interface actually prevent NEC/NEC oligomerization; and 3) no evidence that a hexagonal lattice forms during membrane budding in vitro.

2) Statements on lines 27-28 “….that EBV NEC oligomerizes and that the oligomeric interface is conserved across homologs and essential for budding.” is not supported by the data. This also applies to statements/conclusions on lines 123-124, 354-356 and 385-386.

Reviewer #2: None required

Reviewer #3: (No Response)

**Part III – Minor Issues: Editorial and Data Presentation Modifications**

Reviewer #1: Minor concerns

1) Line 38. In addition to mammals and birds, ICTV indicates that viruses infecting turtles and iguanas are also included within the Herpesviridae.

2) Line 69. Not strictly true. Rabbit skin cells support the replication of HSV-1 UL31 mutants fairly well (PMID: 15476875).

3) Line 101. Are authors are referring to “primary envelopment of capsids” rather than “nuclear egress”? Nuclear egress comprises the entirety of capsid recruitment to the inner nuclear membrane through de-envelopment of primary enveloped virions at the outer nuclear membrane.

4) Figure 4. Consider using a color other than yellow to highlight the tllt angles. The yellow is difficult to see.

5) Figure 5. Consider adding an example of what the budding assays look like (i.e. actual data) in addition to the table. Is there any evidence of a hexagonal lattice of NEC components on these membranes as described in Bigalke et al (PMID: 24916797)?

Reviewer #2: (No Response)

Reviewer #3: - Given the relatively low resolution of the structures, how good is the overall fitting of the models for the 5 NEC copies? How many residues are missing in the 5 different copies? How was the modeling of side chains treated throughout the structure relative to the observed electron density? While some of the variability in the different NEC copies is mentioned in the text, it is difficult to get an overview of the heterogeneity in the structures and how these might related to the NEC interfaces. These issues should be addressed either in the main text or methods section and perhaps in a summary supplementary figure.

- Given the relatively low overall conservation of the NECS across herpesviruses, do the authors observe any enrichment of conserved residues at the heterodimer and hexamer/oligomer interfaces?

- The previous structure of the hook-and-groove interaction in the BFLF2/BFRF1 heterodimer has an interface size of 1569Å2, while this interface in the current structures varies in size from 1,425 to 1,704 Å2. What is the effect of this variation on the overall structure of the heterodimer and what regions are varying to accommodate these changes? How does this variation (along with the variations in the globular head domain interactions) contribute to or impact the potential assembly of the EBV NEC heterodimers into the proposed hexameric form? Are some of these distinct conformations more or less compatible with forming hexameric assemblies?

- Lines 314-316: It would be helpful to the reader to briefly explain the budding assay in the main text and also to clarify in Figure 5 what ILVs are. What is the background level of ILVs in this assay relative to the specific signal?

Line 309: typo “added to the C terminus of BFRF1 tether proteins”

Line 622: Reference is made to Figure 4c, showing ILVs, but this figure was not included. I think adding this panel to one of the original figures would be useful for readers unfamiliar with the budding assay.

PLOS authors have the option to publish the peer review history of their article (what does this mean?). If published, this will include your full peer review and any attached files.

Reviewer #1: No

Reviewer #2: No

Reviewer #3: No
---

## [Decision Letter · Decision Letter 1]

17 May 2022

Dear Dr. Heldwein,

Thank you very much for submitting your revised manuscript "The nuclear egress complex of Epstein-Barr virus buds membranes through a distinct, oligomerization-driven mechanism" for consideration at PLOS Pathogens. As with all papers reviewed by the journal, your manuscript was reviewed by members of the editorial board and by several independent reviewers. The reviewers and editors believe that the manuscript has been significantly improved. We are likely to accept this manuscript for publication, providing that you modify the manuscript according to the review recommendations.

Sincerely,

Edward S. Mocarski

Associate Editor

PLOS Pathogens

Urs Greber

Section Editor

PLOS Pathogens

Kasturi Haldar

Editor-in-Chief

PLOS Pathogens

orcid.org/0000-0001-5065-158X

Michael Malim

Editor-in-Chief

PLOS Pathogens

orcid.org/0000-0002-7699-2064

Reviewer Comments (if any, and for reference):

Reviewer's Responses to Questions

**Part I - Summary**

Reviewer #1: This revised manuscript from Thorsen and colleagues is substantially improved over the previous submission and the new cryo-EM experiments, in particular, have led to new conclusions and have resulted in a more interesting study. Problems related to over-interpretation of the findings remain as do a few inaccuracies in the background information provided.

Reviewer #2: The revised manuscript by Thorsen et al., represents a significant improvement over the first. The newer data showing more results and a cryo-EM imaging of the results of the in vitro budding assay are particularly interesting and help raise the impact of the study. This new data, however, has some problematic aspects that are not addressed critically by the authors. Specifically, the cryo-EM images in Figure 7 do not strongly support either the authors’ descriptions of them or the conclusions they attempt to draw.

(i) the “fence-like” appearance of the protein coat in the side view of the EBV NEC budded vesicles can only be apprehended in a small area of the membrane of the budded vesicle shown and bears no resemblance to the side view from the HSV NEC budded vesicle shown. The spacing of the vertical elements in the EBV image is much tighter than that in the HSV image and they are much narrower. I am unconvinced that these vertical elements in the EBV image represent NEC heterodimers or even that they represent the major structural feature of the vesicle side views. It is also worth noting that the clearest structural element in the EBV image shown is a set of lateral striations layered interior to the vesicle membrane on the right hand side of the vesicle, beginning just above the enlarged side view area. It’s not clear why these are not also a focus of analysis. Some quantitation of the frequency (perhaps % of membrane contour that shows these structures) would be helpful and higher resolution images would definitely strengthen their analysis.

(ii) The top views shown do not have the resolution to allow the authors to make any statement about the organization of NEC heterodimers. It’s clear that the organization of EBV and HSV membrane proteins is different in some way, since the HSV top view image suggests some structure and the EBV image really does not. There is, however, no more to say about this. Since hexamers are not evident even in the HSV image shown it is impossible to interpret their absence in the EBV image.

For both these issues, higher resolution images would be very helpful, but at the very least a more critical discussion of the limitations of this data is necessary.

Reviewer #3: The revised manuscript addresses my earlier comments and adds new cryo-EM data that more directly address the oligomeric state of the EBV NEC, indicating that it does not adopt the hexameric form found in other herpesviruses. This data is consistent with the crystallographic structures and indicates plasticity in the assembly of herpesvirus NECS into larger oligomeric lattices on membranes. While the new data do not support the previous conclusion that the EBV NEC forms a conserved hexameric assembly, the mutational and structural studies support the conclusion that herpesvirus NECS utilize similar interfaces to mediate budding.

**Part II – Major Issues: Key Experiments Required for Acceptance**

Reviewer #1: Major concerns:

1) Line 426 and line 564 of the “clean” version of the MS. Despite the conclusions of these section headings, the authors have not demonstrated that NEC oligomerization is “essential” or “required” for budding and these statements represent an over-interpretation of the findings. The data clearly demonstrate NEC oligomerization occurs and that some mutations (but not others) introduced into the NEC/NEC interface that were intended to prevent oligomerization interfered with budding, however, NEC/NEC oligomerization of these mutants was not assessed. Thus, the authors cannot conclude that oligomerization is essential, or required, for budding. The authors can conclude that residues in the NEC/NEC oligomerization interface are required for optimal budding.

Reviewer #2: No experiments are absolutely required.

Reviewer #3: (No Response)

**Part III – Minor Issues: Editorial and Data Presentation Modifications**

Reviewer #1: Minor concerns:

1) Lines 34-35 of the “clean” version of the MS. "nuclear budding" is not the first step in the process of nuclear egress. At the very least, budding/capsid envelopment is preceded by recruitment of capsids to the inner nuclear membrane as is activation of the NEC, which has been shown, by several groups, to be negatively regulated in infected cells.

2) Lines 56-58 of the “clean” version of the MS. This statement is misleading - e.g. it implies that KSHV infects "most of the world's population", which is not true.

3) Line 788. Dr. Draganova is an author of the revised manuscript and likely does not require additional acknowledgement in the Acknowledgements section.

Reviewer #2: The paper is well written and data are clearly presented. Higher resolution images for Figure 7 would allow stronger conclusions.

Reviewer #3: (No Response)

PLOS authors have the option to publish the peer review history of their article (what does this mean?). If published, this will include your full peer review and any attached files.

Reviewer #1: No

Reviewer #2: No

Reviewer #3: No

Figure Files:

Data Requirements:

Reproducibility:

References:

---

## [Editor Report · Decision Letter 2]

28 May 2022

Dear Dr. Heldwein,

We are pleased to inform you that your manuscript 'The nuclear egress complex of Epstein-Barr virus buds membranes through an oligomerization-driven mechanism' has been provisionally accepted for publication in PLOS Pathogens.

Best regards,

Edward S. Mocarski

Associate Editor

PLOS Pathogens

Urs Greber

Section Editor

PLOS Pathogens

Kasturi Haldar

Editor-in-Chief

PLOS Pathogens

orcid.org/0000-0001-5065-158X

Michael Malim

Editor-in-Chief

PLOS Pathogens

orcid.org/0000-0002-7699-2064
---

## [Editor Report · Acceptance letter]

6 Jul 2022

Dear Dr. Heldwein,

We are delighted to inform you that your manuscript, "The nuclear egress complex of Epstein-Barr virus buds membranes through an oligomerization-driven mechanism," has been formally accepted for publication in PLOS Pathogens.

Best regards,

Kasturi Haldar

Editor-in-Chief

PLOS Pathogens

orcid.org/0000-0001-5065-158X

Michael Malim

Editor-in-Chief

PLOS Pathogens

orcid.org/0000-0002-7699-2064